# Interpolating amplitudes

**Víctor Bresó-Pla,**[1*] **Gudrun Heinrich,**[2†] **Vitaly Magerya,**[3‡] **and Anton Olsson**[2∘]

**1** Institute for Theoretical Physics, University of Heidelberg, 69120 Heidelberg, Germany
**2** Institute for Theoretical Physics, Karlsruhe Institute of Technology, 76131 Karlsruhe, Germany
**3** Theoretical Physics Department, CERN, 1211 Geneva 23, Switzerland

* v.breso@thphys.uni-heidelberg.de † gudrun.heinrich@kit.edu
‡ vitaly.magerya@cern.ch ∘ anton.olsson@kit.edu

## Abstract

The calculation of scattering amplitudes at higher orders in perturbation theory has reached a high degree of maturity. However, their usage to produce physical predictions within Monte Carlo programs is often precluded by the slow evaluation of two- and higher-loop virtual amplitudes, particularly those calculated numerically. As a remedy, interpolation frameworks have been successfully used for amplitudes depending on up to two kinematic invariants. For amplitude interpolation with more variables, such as the five dimensions of a $2 \to 3$ phase space, efficient and reliable solutions are sparse.

This work aims to pave the way for using amplitude interpolation in higher-dimensional phase spaces by reviewing state-of-the-art interpolation methods, and assessing their performance on a selection of $2 \to 3$ scattering amplitudes. Specifically, we investigate interpolation methods based on polynomials, splines, spatially adaptive sparse grids, and neural networks (multilayer perceptron and Lorentz-Equivariant Geometric Algebra Transformer), all under the constraint of limited obtainable data. Our additional aim is to motivate further studies of the interpolation of scattering amplitudes among both physicists and mathematicians.

# 1 Introduction

Particle physics today demands high-precision calculations of scattering amplitudes to find hints of physics beyond the Standard Model (BSM) in the data from current colliders [1, 2]. Even more precision is needed to ensure an adequate exploitation of the much more precise data expected from future colliders [3, 4].

Impressive progress has been made in the calculation of QCD corrections at next-to-next-to-leading order (NNLO) and beyond to the most important processes at the Large Hadron Collider (LHC) [5–7]. However, most of those calculations are restricted to processes involving relatively few kinematic scales or masses, since analytic calculations of the required two-loop amplitudes become rapidly unfeasible as the number of external legs and/or particle masses increases. This stems from the fact that the classes of analytic functions representing the result get more complicated, and from the sheer algebraic complexity. In contrast, numerical methods exhibit a more tractable growth in complexity with the number of scales, see e.g. section 3 of Ref. [5] and Refs. [8–11]. Therefore, the domains of two-loop amplitudes beyond $2 \to 2$ scattering, involving five or more kinematic scales, as well as the domains of multi-loop corrections in electro-weak or BSM theories involving many different masses, are the ones where numerical methods can prove extremely useful. On the other hand, the usefulness in phenomenological applications is strongly tied to the speed and accuracy at which these amplitudes can be evaluated. In a realistic Monte Carlo evaluation of total or differential cross sections, typically millions of phase-space points need to be evaluated [12, 13]. Under these conditions, it is unfeasible to carry out the lengthy numerical evaluation of a two-loop amplitude for each individual phase-space point.

Therefore, a promising way to make numerical methods for multi-loop amplitudes more practical is to precompute the values of the amplitudes at some set of points, often a grid, and rely on interpolation to evaluate them at other phase-space points, i.e., for events produced by the Monte Carlo program. While a lot of progress has been achieved in speeding up tree-level and NLO event generation, see e.g. Refs. [14–17], our final goal is to make multi-loop amplitudes that are not known in analytic form, but have been evaluated numerically, accessible for phenomenology. The present work is a first step towards this goal.

Procedures based on grids have been used successfully for several two-loop amplitudes already, such as the two-loop amplitudes entering $t\bar{t}$-production at NNLO [18,19], Higgs boson pair production in gluon fusion at NLO [20,21] or Higgs+jet production at NLO [22,23]. The two-loop amplitudes entering these $2 \to 2$ processes depend on just two kinematic variables once the masses have been fixed to their Standard Model values. In contrast, the interpolation for $2 \to 3$ processes needs to be performed on a five-dimensional phase space, which makes achieving precision a highly non-trivial task.

The intention of this work is to present a selection of state-of-the-art multidimensional interpolation methods and study their practical usage and performance, having the physics use case in mind. While there is vast mathematical literature on interpolation, the most commonly used function spaces (such as Sobolev and Korobov spaces) are quite general and do not necessarily capture the specific properties of functions related to loop amplitudes. The purpose of this article is to connect the mathematical literature with the physics use cases that so far are mostly limited to interpolation in dimensions lower than five.

This paper is structured as follows. In Section 2 we give a brief introduction to amplitudes and cross sections. We formulate the interpolation problem and define the error metrics and test functions we use to benchmark the approximation methods. In Sections 3–6, four categories of methods are described. Section 3 is about polynomial interpolation, Section 4 about B-splines, Section 5 about sparse grids and Section 6 about machine learning techniques. In these sections, the main results are presented in the form of plots of the approximation error

against the amount of training data. We conclude in Section 7 by summarising the results of the previous sections. Further details on the parametrization of the test functions are given in Appendix A.

Implementations of the described methods, as well as the test functions used for the benchmarks, can be found at https://github.com/OlssonA/interpolating_amplitudes.

## 2  Setting the stage

### 2.1  What is an amplitude

In a proton–proton collider of collision energy $\sqrt{s}$, the number of times a reaction

$$\underbrace{p + p}_{\text{protons}} \to \underbrace{a(q_1) + b(q_2)}_{\text{partons}} \to \underbrace{c(p_1) + d(p_2) + \ldots}_{\text{particles}} \tag{1}$$

takes place is proportional to the *differential cross section* of the process [24], which, assuming that the squared masses of partons $a$ and $b$ are much smaller than $s$, is

$$\mathrm{d}\sigma = \frac{1}{2\hat{s}} |\mathcal{M}(q_1, q_2, p_1, p_2, \ldots)|^2 \, \mathrm{d}\Phi \, \mathrm{d}\rho_{a,b}(\hat{s}, s), \tag{2}$$

where $\mathcal{M}$ is the *scattering amplitude* for the process $a + b \to c + d + \ldots$, $\mathrm{d}\Phi$ is the element of the Lorentz-invariant phase space, given in terms of the four-momenta $q_i$, $p_i$, and masses $m_i$ as

$$\mathrm{d}\Phi = (2\pi)^4 \, \delta^4(q_1 + q_2 - \sum_i p_i) \prod_i \frac{\mathrm{d}^4 p_i}{(2\pi)^4} 2\pi \delta(p_i^2 - m_i^2) \, \theta(p_i^{(0)}), \tag{3}$$

and $\mathrm{d}\rho$ is the probability of finding partons $a$ and $b$ with collision energy $\sqrt{\hat{s}}$ in the colliding proton pair, given via the *parton distribution functions* $f_a, f_b$ as

$$\mathrm{d}\rho_{a,b}(\hat{s}, s, \mu_F) = \mathrm{d}\hat{s} \int f_a(x_1, \mu_F) f_b(x_2, \mu_F) \, \delta(\hat{s} - s x_1 x_2) \, \mathrm{d}x_1 \, \mathrm{d}x_2. \tag{4}$$

Parton distribution functions themselves are well studied and readily available via [25]. The amplitude is normally calculated within perturbation theory as an expansion in a small coupling parameter, e.g. the strong coupling $\alpha_s$:

$$|\mathcal{M}|^2 \equiv \mathcal{A} = \sum_{k=0}^{\infty} \left( \frac{\alpha_s}{2\pi} \right)^k \mathcal{A}_k. \tag{5}$$

The first term in this expansion is the leading order (LO), the second term is the next-to-leading order (NLO), the third is the next-to-next-to leading order (NNLO), and so on. Each subsequent order is much harder to calculate than the previous one, so when calculating $\mathcal{A}_2$, one can consider $\mathcal{A}_0$ and $\mathcal{A}_1$ to be readily available. Tools exist that largely automate calculation of $\mathcal{A}$ up to NLO [26–30]; beyond that, the calculations are usually custom-made for each process.

For the generic case where the leading order does not contain loop diagrams, NLO cross sections involve one-loop diagrams as well as diagrams with extra radiation that, if unresolved, can lead to soft or collinear singularities, and therefore require a subtraction procedure. At NNLO, the QCD radiation can be doubly unresolved and the subtraction procedures are highly non-trivial. For NNLO processes involving mostly massless particles, for example di-jet production, the double real radiation accounts for the majority of the Monte Carlo integration time in the cross section calculation, while the two-loop virtual diagrams for massless particles are known analytically and are therefore fast to evaluate. The situation changes if more

kinematic scales or masses are entering the virtual corrections, because analytic calculations are either not available or are in a form that does not allow to evaluate them in a Monte Carlo setup. Therefore, we will target loop amplitudes here, aiming at two loops eventually, but using tree-level and one-loop amplitudes as test cases.

The differential cross section depends on the parameters that fully characterize the phase space. These are two for $2 \to 2$ processes, five for $2 \to 3$, nine for $2 \to 4$, etc. There is a freedom to choose these parameters; common choices are energy or Mandelstam variables, $s_{ij} = (p_i + p_j)^2$, and angular variables. In what follows, we denote these parameters as $\vec{x}$, and choose them such that the physical region of the process is a hypercube: $\vec{x} \in (0; 1)^d$.

The primary use of amplitudes is to calculate cross sections—either the total values over the whole phase space: $\sigma \equiv \int d\sigma$, or differential over some phase-space partitioning defined either via some observable quantity $\mathcal{O}$: $d\sigma/d\mathcal{O}$, or via a sequence of them, as in e.g. [31].

## 2.2 Our goal

Assume that we can calculate most parts of a squared amplitude $\mathcal{A}$ quickly and precisely, but one part, e.g. a subleading order in $\alpha_s$, is slow or expensive to evaluate. Let us denote this part as $a : (0; 1)^d \to \mathbb{R}$. We aim to approximate $a$ by some function $\tilde{a}$, from the knowledge of the values of $a$ at some *data points* $\vec{x}_1, \ldots, \vec{x}_n$: $a_i \equiv a(\vec{x}_i)$. We are interested in algorithms to choose $\vec{x}_i$ and to construct $\tilde{a}$ such that it is "close enough" to $a$, while requiring as few data points as possible, as to minimize the expensive evaluations of $a$.

## 2.3 How to define the approximation error

There are different ways to precisely define what "close enough" means, and no single definition works equally well for all observables and phase-space regions of interest, so a choice of what to prioritize must be made.

In this paper we select $d\sigma/d\vec{x}$ as the quantity of interest. We define the approximation error by viewing it as a probability density and using the distance between $d\sigma/d\vec{x}$ based on $\tilde{a}$ and $a$, measured via the $L^1$ norm:[1]

$$\varepsilon = \frac{||\tilde{f} - f||_1}{||f||_1}, \quad \text{where} \quad f(\vec{x}) \equiv a(\vec{x}) \underbrace{\frac{1}{2\hat{s}} \left| \frac{d(\Phi, \rho)}{d\vec{x}} \right|}_{\equiv \text{weight } w(\vec{x})}. \tag{6}$$

The choice of the $L^1$ norm ensures that this quantity is independent of the choice of variables $\vec{x}$ (which would not be the case for e.g. $L^2$ norm). In statistics, $\varepsilon$ is known as the *total variation distance* (up to an overall normalization). This distance weighs different phase-space regions proportional to their contribution to the total cross section (via the factor $w$), and guarantees that for any phase-space subregion $R$, using $\tilde{a}(\vec{x})$ instead of $a(\vec{x})$ will result in the error of $\int_{\vec{x} \in R} d\sigma$ being no more than $\varepsilon \left( \frac{\alpha_s}{2\pi} \right)^k ||f||_1$.

Note that the precision of the total cross section comes at the expense of precision in tails of differential distributions: for parts of the phase space that do not contribute much to the total cross section, such as the very-high-energy region, only low precision is guaranteed by a bound on $\varepsilon$. An alternative definition of the error, $\varepsilon = \max|\tilde{a}/a - 1|$, would ensure equal relative precision for all bins of any differential distribution, but satisfying it would come at the expense of the interpolation spending most effort on phase-space regions where only few (or none at all) experimental data points are expected at the LHC. This is the trade-off involved in error target choices; to apply the methods we study, each application would need to choose its own appropriate error measure.

---

[1] Here and throughout, we define the $L^p$-norm $||f||_p$ as $\left( \int |f(\vec{x})|^p \, d\vec{x} \right)^{1/p}$.

## 2.4 How to prioritize different phase-space regions

Summarizing Section 2.3, we approximate $a(\vec{x})$, and encode the relative importance of different phase-space regions into $w(\vec{x})$. But can we incorporate this importance information to improve the approximation procedure? There are multiple options:

1. Instead of constructing an approximation for $a(\vec{x})$ directly, we can construct an approximation $\widetilde{f}(\vec{x})$ for $f(\vec{x}) \equiv a(\vec{x})w(\vec{x})$, and then set $\widetilde{a}(\vec{x}) = \widetilde{f}(\vec{x})/w(\vec{x})$.

2. For methods based on regression, we can choose the data points $x_i$ such that they cluster more in regions where $w(\vec{x})$ is greater.

3. For adaptive methods, we can approximate $a(\vec{x})$, but use the quantity $a\,w$ in the adaptivity condition, so that regions where $a\,w$ (and not just $a$) is approximated the worst are refined first.

4. We can try to find a variable transformation from $\vec{x}$ to $\vec{y}$ that cancels $w$, and interpolate in $\vec{y}$ instead of $\vec{x}$. In other words, choose $\tau$ with $\vec{x} = \tau(\vec{y})$ such that $w(\vec{x})\,|\mathrm{d}\tau(\vec{x})/\mathrm{d}\vec{y}| = 1$, so that $\int a\,w\,\mathrm{d}x = \int a\,\mathrm{d}y$.

   One popular approach to construct such transformations is to approximate $\tau$ as its rank-1 decomposition: $\tau(\vec{y}) = \prod_i \tau_i(y_i)$. This is the basis of the VEGAS integration algorithm [32]. Higher-rank approximations are, of course, also possible. Another approach is based on *normalizing flows* [33]: it involves fitting a Jacobian of a specially-constructed transformation to $1/w$. This machine learning approach has been extensively used to develop improvements in importance sampling [14, 16, 34–36].

   These approaches can be applied on top of any interpolation method, and we shall not discuss them further.[2]

We study these approaches for each of the interpolation methods considered in this paper.

## 2.5 The test functions

In the remainder of the article we assess the performance of the most promising interpolation algorithms studied in numerical analysis when applied to the following five test functions.

*Test function $f_1$:*

Leading order amplitude for $q\bar{q} \to t\bar{t}H$ taken as a 5-dimensional function over the phase space as described in [37]. Specifically,

$$a_1 = \langle \mathcal{M}_0^{q\bar{q}t\bar{t}H}|\mathcal{M}_0^{q\bar{q}t\bar{t}H}\rangle, \quad f_1 = a_1 \times \left|\frac{\mathrm{d}\Phi_{t\bar{t}H}}{\mathrm{d}(\mathrm{frac}_{s_{t\bar{t}}}, \theta_H, \theta_t, \varphi_t)}\right| \times \frac{1}{2\hat{s}}\frac{\mathrm{d}\rho_{q\bar{q}}}{\mathrm{d}\beta^2} \times J_{t\bar{t}H}, \quad (7)$$

with the phase space parameters set as

$$\beta^2 = \frac{10}{100} + \frac{86}{100}x_1, \quad \mathrm{frac}_{s_{t\bar{t}}} = x_2, \quad \theta_H = \pi x_3, \quad \theta_t = \pi x_4, \quad \varphi_t = 2\pi x_5, \quad (8)$$

$$J_{t\bar{t}H} = \left|\frac{\mathrm{d}(\beta^2, \mathrm{frac}_{s_{t\bar{t}}}, \theta_H, \theta_t, \varphi_t)}{\mathrm{d}\vec{x}}\right| = \frac{86}{50}\pi^3. \quad (9)$$

---

[2]In our very limited testing, we did not see major gains from using these.

The phase space parameters are described in more detail in Appendix A.1. To specify $\rho_{q\bar{q}}$ it is possible to use eq. (4) together with one of the well known parton distribution functions, but to be more self-contained, we use the following generic form of $\rho$:

$$\frac{1}{2\hat{s}}\frac{\mathrm{d}\rho_{q\bar{q}}}{\mathrm{d}\beta^2} \propto \frac{(1-c_1\beta^2)^2}{(1-c_2\beta^2)(1-c_3\beta^2)}, \tag{10}$$

and set $c = \{1.0132, 0.9943, 0.3506\}$, which approximates $\rho$ computed using the parton distribution functions from [38]. The overall proportionality constant is omitted here because it cancels in the error definition of eq. (6).

The amplitude $a_1$ possesses multiple discrete symmetries. The most interesting ones are the symmetries under $\varphi_t \to -\varphi_t$ (parity invariance), and the swap $\{q, t\} \leftrightarrow \{\bar{q}, \bar{t}\}$. These two symmetries are present in all higher order corrections to this amplitude too; the leading order is additionally symmetric under the swap $q \leftrightarrow \bar{q}$. In terms of the variables $x_i$, this works out to

$$\varphi_t \to -\varphi_t : \qquad a_1(x_5) = a_1(1-x_5), \tag{11}$$
$$\{q, t\} \leftrightarrow \{\bar{q}, \bar{t}\} : \qquad a_1(x_3, x_4) = a_1(1-x_3, 1-x_4), \tag{12}$$
$$q \leftrightarrow \bar{q} : \qquad a_1(x_3, x_5) = a_1(1-x_3, x_5 + 1/2). \tag{13}$$

As an illustration, slices of the amplitude $a_1$ in $x_1$–$x_2$ and $x_1$–$x_4$ space are as follows:

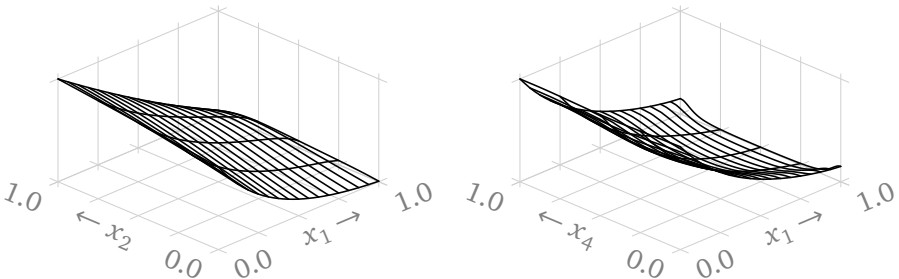

The same slices for the function $f_1$ are:

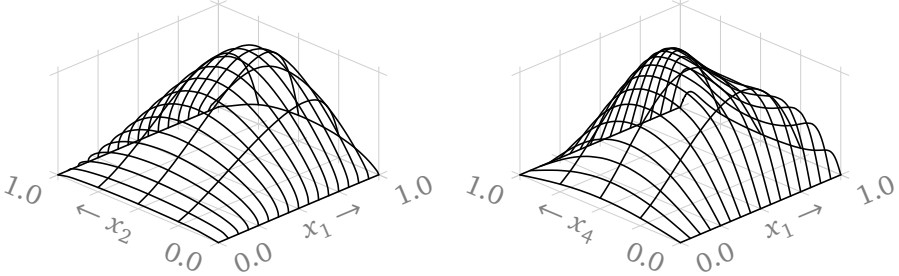

*Test function $f_2$:*

One-loop amplitude contributing to $q\bar{q} \to t\bar{t}H$, taken as a 5-dimensional function. This amplitude has an integrable singularity at $\mathrm{frac}_{s_{t\bar{t}}} \to 0$ due to the exchange of a gluon between two top-quarks that can have very low energy (the configuration approaches a so-called "Coulomb-singularity"), which needs to be tamed for interpolation to work well. We do this by subtracting the singular behaviour using eq. (2.67) from [39], i.e.:

$$a_2 = 2\,\mathrm{Re}\left[\langle \mathcal{M}_0^{q\bar{q}t\bar{t}H} | \mathcal{M}_1^{q\bar{q}t\bar{t}H} \rangle + \frac{\pi^2}{\beta_{t\bar{t}}}\langle \mathcal{M}_0^{q\bar{q}t\bar{t}H} | \mathbf{T}_{t\bar{t}} | \mathcal{M}_0^{q\bar{q}t\bar{t}H} \rangle \right], \tag{14}$$

$$f_2 = a_2 \times \left| \frac{\mathrm{d}\Phi_{t\bar{t}H}}{\mathrm{d}(\mathrm{frac}_{s_{t\bar{t}}}, \theta_H, \theta_t, \varphi_t)} \right| \times \frac{1}{2\hat{s}}\frac{\mathrm{d}\rho_{q\bar{q}}}{\mathrm{d}\beta^2} \times J_{t\bar{t}H}, \tag{15}$$

where $\mathbf{T}_{t\bar{t}}$ is a colour operator defined in [39] and $\beta_{t\bar{t}}$ is the velocity of the $t\bar{t}$ system, in our variables given by

$$\beta_{t\bar{t}}^2 \equiv 1 - \left[ 1 + \text{frac}_{s_{t\bar{t}}} \left( \left( \frac{1 + \frac{1}{2}\frac{m_H}{m_t}}{\sqrt{1 - \beta^2}} - \frac{1}{2}\frac{m_H}{m_t} \right)^2 - 1 \right) \right]^{-1}. \tag{16}$$

Both the phase space and $\rho_{q\bar{q}}$ are the same as for $f_1$. The amplitude $a_2$ possesses two of the symmetries of $a_1$:

$$a_2(x_5) = a_2(1 - x_5), \quad \text{and} \quad a_2(x_3, x_4) = a_2(1 - x_3, 1 - x_4). \tag{17}$$

Slices of the amplitude $a_2$ in $x_1$–$x_2$ and $x_1$–$x_4$ space are as follows:

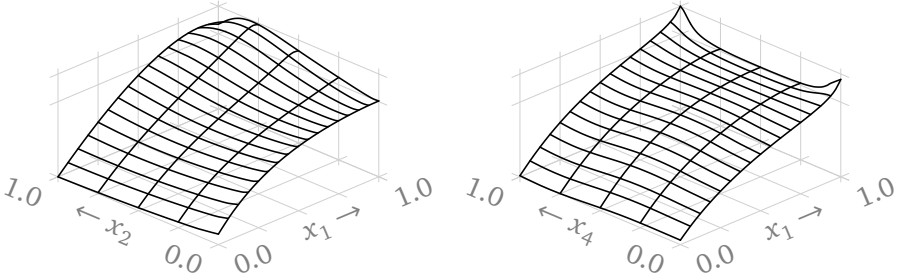

The same slices for the function $f_2$ are:

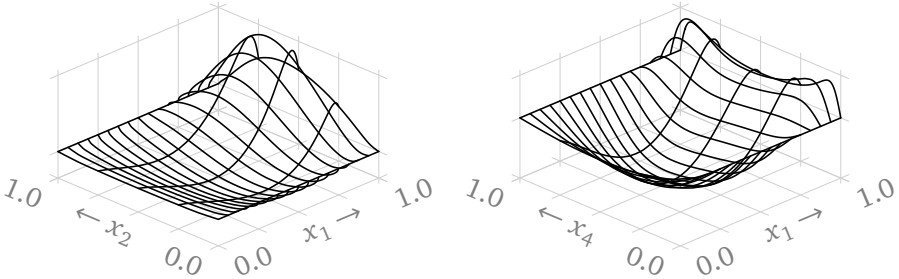

*Test function $f_3$:*

Leading order amplitude for $gg \to t\bar{t}H$,

$$a_3 = \langle \mathcal{M}_0^{ggt\bar{t}H} | \mathcal{M}_0^{ggt\bar{t}H} \rangle, \quad f_3 = a_3 \times \left| \frac{\mathrm{d}\Phi_{t\bar{t}H}}{\mathrm{d}(\text{frac}_{s_{t\bar{t}}}, \theta_H, \theta_t, \varphi_t)} \right| \times \frac{1}{2\hat{s}} \frac{\mathrm{d}\rho_{gg}}{\mathrm{d}\beta^2} \times J_{t\bar{t}H}, \tag{18}$$

with the same kinematics and phase space as for $f_1$, and $\mathrm{d}\rho_{gg}/\mathrm{d}\beta^2$ chosen via the ansatz of eq. (10) with $c = \{1.0134, 0.7344, 0.0987\}$.

The amplitude $a_3$ possesses the same symmetries as $a_1$, given in eq. (13).

Slices of the amplitude $a_3$ in $x_1$–$x_2$ and $x_1$–$x_4$ space are as follows:

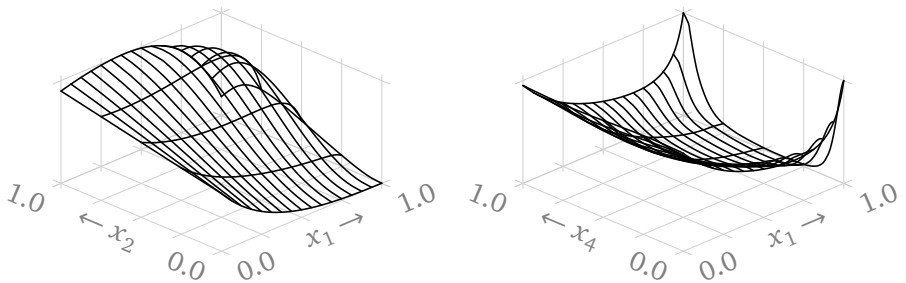

The same slices for the function $f_3$ are:

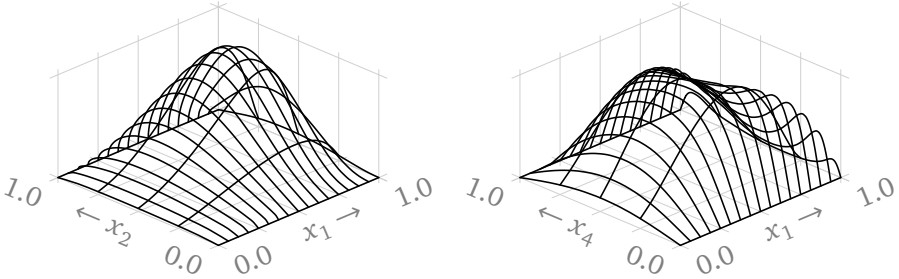

*Test function $f_4$:*

One-loop amplitude contributing to $gg \to t\bar{t}H$, taken as a 5-dimensional function, with the Coulomb-type singularity subtracted in the same way as for $a_2$, i.e.,

$$a_4 = 2\mathrm{Re}\left[ \langle \mathcal{M}_0^{ggt\bar{t}H} | \mathcal{M}_1^{ggt\bar{t}H} \rangle + \frac{\pi^2}{\beta_{t\bar{t}}} \langle \mathcal{M}_0^{ggt\bar{t}H} | \mathbf{T}_{t\bar{t}} | \mathcal{M}_0^{ggt\bar{t}H} \rangle \right], \tag{19}$$

$$f_4 = a_4 \times \left| \frac{\mathrm{d}\Phi_{t\bar{t}H}}{\mathrm{d}(\mathrm{frac}_{s_{t\bar{t}}}, \theta_H, \theta_t, \varphi_t)} \right| \times \frac{1}{2\hat{s}} \frac{\mathrm{d}\rho_{gg}}{\mathrm{d}\beta^2} \times J_{t\bar{t}H}, \tag{20}$$

where $\beta_{t\bar{t}}$ is given in eq. (16), and the rest is the same as for $f_3$.

The amplitude $a_4$ possesses the same symmetries as $a_3$ and $a_1$, given in eq. (13).

Slices of the amplitude $a_4$ in $x_1$–$x_2$ and $x_1$–$x_4$ space are as follows:

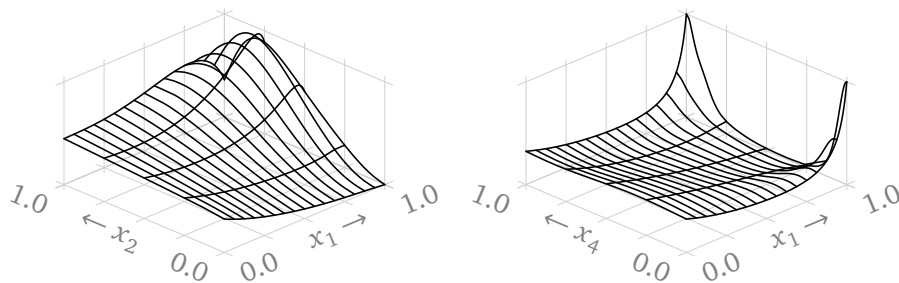

The same slices for the function $f_4$ are:

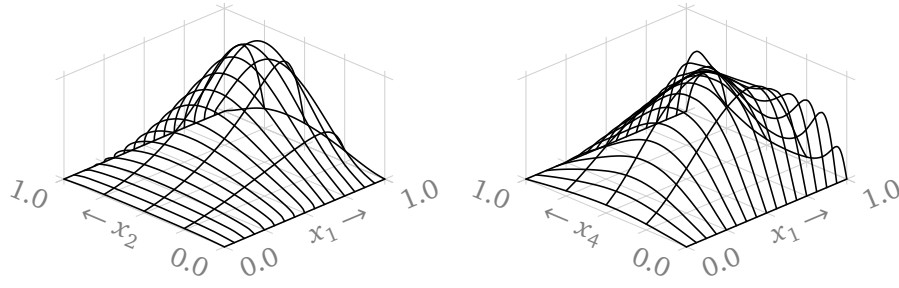

*Test function $f_5$:*

Leading order (one-loop) amplitude for $gg \to Hg$, taken as a 2-dimensional function,

$$a_5 = \langle \mathcal{M}_1^{ggHg} | \mathcal{M}_1^{ggHg} \rangle, \quad f_5 = a_5 \times \frac{\mathrm{d}\Phi_{Hg}}{\mathrm{d}\theta_H} \times \frac{1}{2\hat{s}} \frac{\mathrm{d}\rho_{gg}}{\mathrm{d}\beta^2} \times J_{Hg}, \tag{21}$$

with the phase-space parameters set as

$$\beta^2 = \frac{33}{100} + \frac{66}{100}x_1, \quad \theta_H = \theta_0 + (\pi - 2\theta_0)x_2, \tag{22}$$

$$J_{Hg} = \left| \frac{\mathrm{d}(\beta^2, \theta_H)}{\mathrm{d}\vec{x}} \right| = \frac{66}{100}(\pi - 2\theta_0), \tag{23}$$

the phase-space density being

$$\frac{\mathrm{d}\Phi_{Hg}}{\mathrm{d}\theta_H} = \frac{1}{16\pi}\frac{1}{\hat{s}}\sqrt{\lambda(\hat{s}, m_H^2, 0)}\sin\theta_H = \frac{\beta^2\sin\theta_H}{16\pi}, \tag{24}$$

and $\mathrm{d}\rho_{gg}/\mathrm{d}\beta^2$ chosen via the ansatz of eq. (10), with $c = \{1.0012, 0.9802, 0.3357\}$.

The introduction of $\theta_0$ as a cutoff is needed because $a_5$ diverges as $1/\sin^2\theta_H$ at $\theta_H \to 0$ and $\theta_H \to \pi$. We choose not to interpolate the region around the divergence, because in practical calculations it should be regulated by infrared subtraction or appropriate kinematic cuts; we only consider the phase-space region where the transverse momentum $p_T$ of the Higgs boson $H$ is greater than a cutoff $p_{T,0}$. Then:

$$\sin\theta_0 = p_{T,0}\frac{2\sqrt{\hat{s}}}{\sqrt{\lambda(\hat{s}, m_H^2, 0)}} = 2\frac{p_{T,0}}{m_H}\frac{\sqrt{1 - \beta^2}}{\beta^2}, \tag{25}$$

We choose $p_{T,0}$ corresponding to the lower boundary of the $\beta^2$ region from eq. (22):

$$\frac{p_{T,0}}{m_H} = \frac{1}{2}\frac{\beta_{\min}^2}{\sqrt{1 - \beta_{\min}^2}}, \tag{26}$$

such that at $\beta^2 < \beta_{\min}^2$ no phase space point passes the $p_T$ cut. This works out to $p_{T,0} \approx 25$ GeV.

The amplitude $a_5$ is symmetric under the swap of the incoming gluons, i.e.,

$$a_5(x_2) = a_5(1 - x_2). \tag{27}$$

The amplitude $a_5$ (left) and the function $f_5$ (right) depend on $x_1$ and $x_2$ as follows:

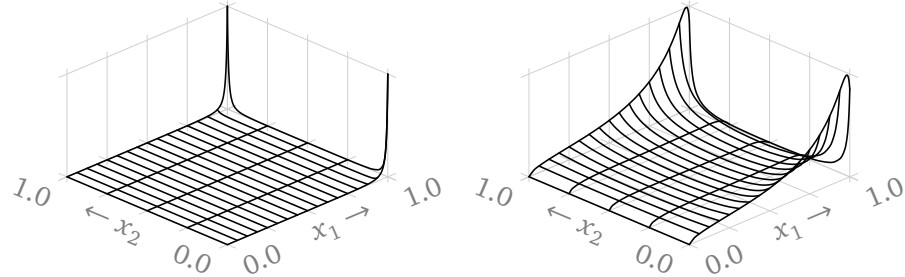

In all cases we use GOSAM [29] to evaluate the amplitudes, and set $m_H^2/m_t^2 = 12/23$.

## 2.6 How to use test function symmetries

When the test functions are symmetric under discrete transformations, as ours are, there are multiple ways to take advantage of this:

1. Make the interpolant obey the same symmetries by construction.

2. Duplicate symmetric data points, i.e., if $f(x) = f(1-x)$, then for each $x_i$ also add $1-x_i$ to the data set (but still count this as a single evaluation of $f$).

3. Reduce the interpolation domain, i.e., if $f(x) = f(1-x)$, only construct the approximation for $x \in [0, 1/2]$, and use the symmetry to obtain the values in $x \in [1/2, 1]$.

Each of our interpolation methods makes use of at least one of these techniques to enforce symmetry awareness.

## 2.7 How to evaluate the approximation error

To evaluate eq. (6) we use Monte Carlo integration. It can be formulated based on different kinds of testing samples:

$$\text{uniform:} \qquad \vec{x}_1, \ldots \vec{x}_m \sim 1, \qquad \varepsilon = \frac{\sum_{i=1}^m |(\widetilde{a}_i - a_i) w_i|}{\sum_{i=1}^m |a_i w_i|}, \qquad (28)$$

$$\text{partially unweighted:} \qquad \vec{x}_1, \ldots, \vec{x}_m \sim w, \qquad \varepsilon = \frac{\sum_{i=1}^m |\widetilde{a}_i - a_i|}{\sum_{i=1}^m |a_i|}, \qquad (29)$$

$$\text{leading-order unweighted:} \quad \vec{x}_1, \ldots, \vec{x}_m \sim a_{\text{LO}} w, \quad \varepsilon = \frac{\sum_{i=1}^m |(\widetilde{a}_i - a_i)/a_{\text{LO},i}|}{\sum_{i=1}^m |a_i/a_{\text{LO},i}|}. \qquad (30)$$

The first of these is natural for the selected variables $\vec{x}$, the second is important because it is independent of the choice of $\vec{x}$ (and can be approximated by e.g. the RAMBO sampling technique [40]), and the third is important because it is oriented at the approximate probability distribution of physical scattering events, encoded in the leading-order amplitude, $a_{\text{LO}}$.

While each sampling method must yield the same result asymptotically, in practice we are interested in using as few testing evaluations as possible. In Figure 1 we compare the error convergence when using the different sampling methods for test function $f_2$. Here we see that the error becomes stable for all methods after $m \sim 1000$ samples.

Note that a uniform sample can be generated both via Monte Carlo, i.e., randomly, and with a low-discrepancy sequence (such as the Sobol sequence). In Figure 1 we present results for both options. We observe that, even though sampling from a low-discrepancy sequence gives a more uniform coverage of the parameter space, we see only marginal improvements in the error convergence.

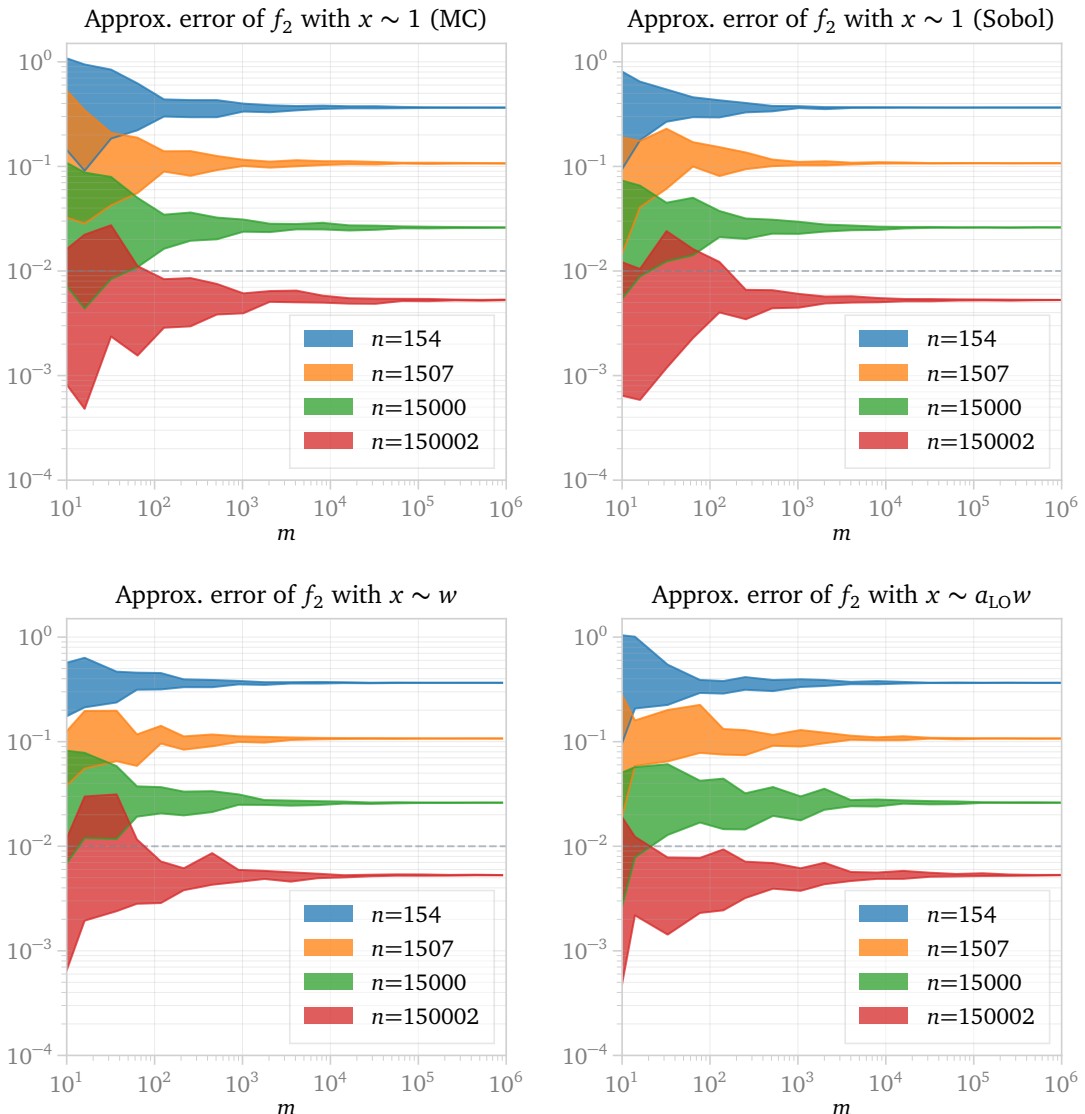

Figure 1: Approximation error $\varepsilon$ of $f_2$, as defined in eq. (6), evaluated via eq. (28), eq. (29), and eq. (30), as a function of the number of testing points $m$. The uniform samples are taken both randomly (MC) and from a low-discrepancy sequence (Sobol). The approximations are constructed using sparse grid interpolation from Section 5, with different numbers of data points $n$. The error bands are created from 10 independent testing sets for each $m$.

## 3  Polynomial interpolation

The classic interpolation method is polynomial interpolation [41, 42]. In the univariate case, $\widetilde{f}$ is constructed as a polynomial of degree $n-1$ that passes through exactly $n$ interpolation nodes $x_i$. It can be written in the *Lagrange form* as

$$\widetilde{f}(x) = \sum_{i=1}^{n} f_i \, l_i(x), \qquad l_i(x) \equiv \prod_{j \neq i} \frac{x - x_j}{x_i - x_j}, \tag{31}$$

or slightly rewritten in the *barycentric form* [43] as

$$\widetilde{f}(x) = \frac{\displaystyle\sum_{i=1}^{n} \frac{w_i}{x - x_i} f_i}{\displaystyle\sum_{i=1}^{n} \frac{w_i}{x - x_i}}, \qquad w_i = \prod_{j \neq i} \frac{1}{x_i - x_j}. \tag{32}$$

The barycentric interpolation formula is general enough that any *rational* interpolation can be expressed by it through an appropriate choice of the *weights* $w_i$, ; the weights given here, however, correspond to the purely polynomial interpolation of eq. (31).[3] This is our form of choice for evaluation due to its numerical stability and simplicity.

The error of a polynomial approximation is given by

$$f(x) - \widetilde{f}(x) = \frac{f^{(n)}(\xi)}{n!} \prod_{i=1}^{n} (x - x_i), \tag{33}$$

where $\xi$ is some function of $x$. To minimize this error *a priori* without the precise knowledge of $f^{(n)}$, one can choose the nodes $x_i$ such that they would minimize $\prod_i (x - x_i)$ over the domain of interest. Doing so is important because a naive choice of equidistant nodes leads to the *Runge phenomenon* [45]: the higher the degree of the polynomial, the worse the approximation error becomes close to the boundaries. E.g., for 10 nodes on the interval of $[-1; 1]$:

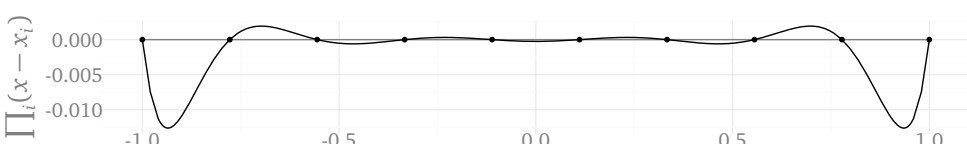

### Chebyshev nodes of the first kind

The product $\prod_i (x - x_i)$ is minimized in the sense of the infinity norm by the *Chebyshev nodes of the first kind*, which are traditionally given for the domain $[-1; 1]$ as

$$x_i = \cos\left(\frac{2i - 1}{2n} \pi\right), \qquad i = 1, \dots, n. \tag{34}$$

These nodes achieve a uniform $\prod_i (x - x_i)$ over the interval:

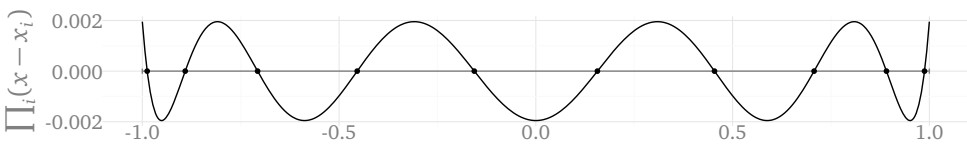

---

[3]Rational interpolation methods, specifically of the non-linear kind, such as the AAA algorithm [44], have established themselves as the most efficient one-dimensional interpolation methods, but generalizations to many dimensions are not developed well enough for us to consider them.

## Chebyshev polynomials

Corresponding to these nodes are the *Chebyshev polynomials of the first kind*:

$$T_k(x) = \cos(k \arccos(x)). \tag{35}$$

Specifically, eq. (34) are the zeros of $T_n(x)$. These polynomials are orthogonal with respect to the weight $1/\sqrt{1-x^2}$:

$$\int_{-1}^{1} \frac{T_n(x)\, T_m(x)}{\sqrt{1-x^2}}\, \mathrm{d}x = \begin{cases} \pi & \text{if } n = m = 0, \\ \pi/2 & \text{if } n = m \neq 0, \\ 0 & \text{if } n \neq m. \end{cases} \tag{36}$$

Note that the nodes in eq. (34) are nothing more than equidistant points in $\phi = \arccos(x)$, and the corresponding polynomials are simply an even Fourier series in $\phi$. This is why a transformation from the function values $\{f_i\}$ to the coefficients of the decomposition into Chebyshev polynomials $\{c_i\}$,

$$\widetilde{f}(x) = \sum_i c_i\, T_i(x), \tag{37}$$

is just a Fourier transform (specifically, a discrete cosine transform). Still, for interpolation purposes, the barycentric form of eq. (32) is preferable, since both eq. (37) and the monomial form suffer from rounding errors that prevent their usage for $n \gtrsim 40$.

## Chebyshev nodes of the second kind

A related set of points that avoids the Runge phenomenon is *Chebyshev nodes of the second kind* (a.k.a. *Chebyshev–Lobatto nodes*), traditionally given as

$$x_i = \cos\left(\frac{i-1}{n-1}\pi\right), \qquad i = 1, \ldots, n. \tag{38}$$

Unlike eq. (34), these points are not located at the zeros of $T_n(x)$, but rather at the extrema and end points, with the advantage of them being nested: the set of $n$ of these is exactly contained in the set of $2n-1$. This comes at the price of $\prod_i(x-x_i)$ not being uniform over the interval:

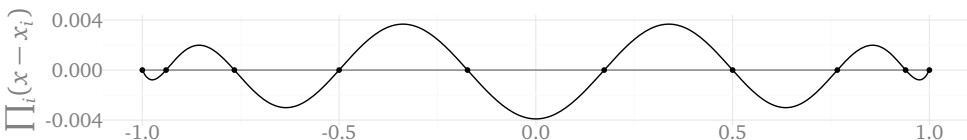

The property of being nested is important for adaptive interpolation constructions because larger grids can reuse the results of the smaller ones that they contain.

Unfortunately, the inclusion of the end points can make this construction impractical for scattering amplitudes. For example, $f_2$ can not be evaluated at exactly $x_2 = 0$ due to a loss of numerical precision, and evaluation at $x_1 = 1$ is possible but undesirable because evaluation time typically grows when approaching this boundary.

## Gauss nodes

Closely related to Chebyshev nodes, and often considered superior, are *Gauss nodes* and *Gauss–Lobatto nodes*. These are defined respectively as the location of zeros and extrema of the Legendre polynomials. They have the advantage that a quadrature built on them (*Gauss quadrature*)

is exact for polynomials up to degree $2n - 1$, while the same for Chebyshev nodes (*Clenshaw–Curtis quadrature*) is only exact for polynomials up to degree $n - 1$. In practice, however, the approximation error of both is very close [46, 47], and since Gauss nodes are much harder to compute compared to eq. (34), we do not consider them further.

## 3.1 Approximation error scaling

It is known that polynomial interpolation at Chebyshev nodes is logarithmically close to the best polynomial interpolation of the same degree [41]. The approximation error itself depends on how smooth the function $f$ is [42, 48, 49]. If $f$ has $\nu - 1$ continuous derivatives and the variation of $f^{(\nu)}$ is bounded, then

$$||f - \widetilde{f}||_2 \leq \frac{4\,||f^{(\nu)}||_1}{\pi\,\nu(n - \nu)^\nu}, \quad \text{for } n > \nu. \tag{39}$$

If $f$ is analytic, and can be analytically continued to an ellipse in the complex plane with focal points at $\pm 1$ and the sum of semimajor and semiminor axes $\rho$ (a *Bernstein ellipse*), then

$$||f - \widetilde{f}||_2 \leq \frac{4M\rho^{-n}}{\rho - 1}, \quad \text{where } M = \max|f(x)| \text{ in the ellipse.} \tag{40}$$

## 3.2 Multiple dimensions

The simplest generalization to multiple dimensions is to take the set of nodes $\{\vec{x}_i\}$ to be the outer tensor product of the Chebyshev nodes of eq. (34) for each dimension,

$$\{\vec{x}_i\} = \{x_{i_1}\} \otimes \cdots \otimes \{x_{i_d}\}, \tag{41}$$

with possibly different node count in each dimension, $n_i$. This corresponds to interpolation via nested application of eq. (32), or via the decomposition

$$\widetilde{f}(\vec{x}) = \sum_{i_1=1}^{n_1} \cdots \sum_{i_d=1}^{n_d} c_{i_1 \ldots i_d}\, T_{i_1}(x_1) \cdots T_{i_d}(x_d). \tag{42}$$

A detailed study of the interpolation error of this construction is presented in [50]. Roughly speaking, it is similar to eq. (39) and eq. (40), except instead of $n$ one must use $n_i$, which are of the order of $\sqrt[d]{n}$, leading to progressively slower convergence as $d$ increases. This is known as *the curse of dimensionality* [51].

## 3.3 Dimensionally adaptive grid

The tensor product construction is fairly rigid in that it allows for no local refinement; only the per-dimension node counts $n_i$ can be tuned. Such tuning is sometimes referred to as *dimensional adaptivity*, and it can be beneficial. To examine that, let us inspect the coefficients $c_{i_1 \ldots i_5}$ corresponding to $f_1$: their two-dimensional slices are presented in Figure 2. From these we can learn that the function has discrete symmetries in the 3-rd, 4-th, and 5-th dimensions that make a subset of the basis coefficients zero, forming a checkerboard pattern. We also learn that the coefficients decay much faster in the 5th dimension compared to the rest.

To make use of this insight, let us study Figure 3, where the scaling of $c$ is depicted in each dimension. If we want to sample so that coefficients in each direction are close to each other (to prevent oversampling), we would need to maintain a ratio of, e.g.,

$$n_1 : n_2 : n_3 : n_4 : n_5 \approx 2 : 4 : 2 : 3 : 1. \tag{43}$$

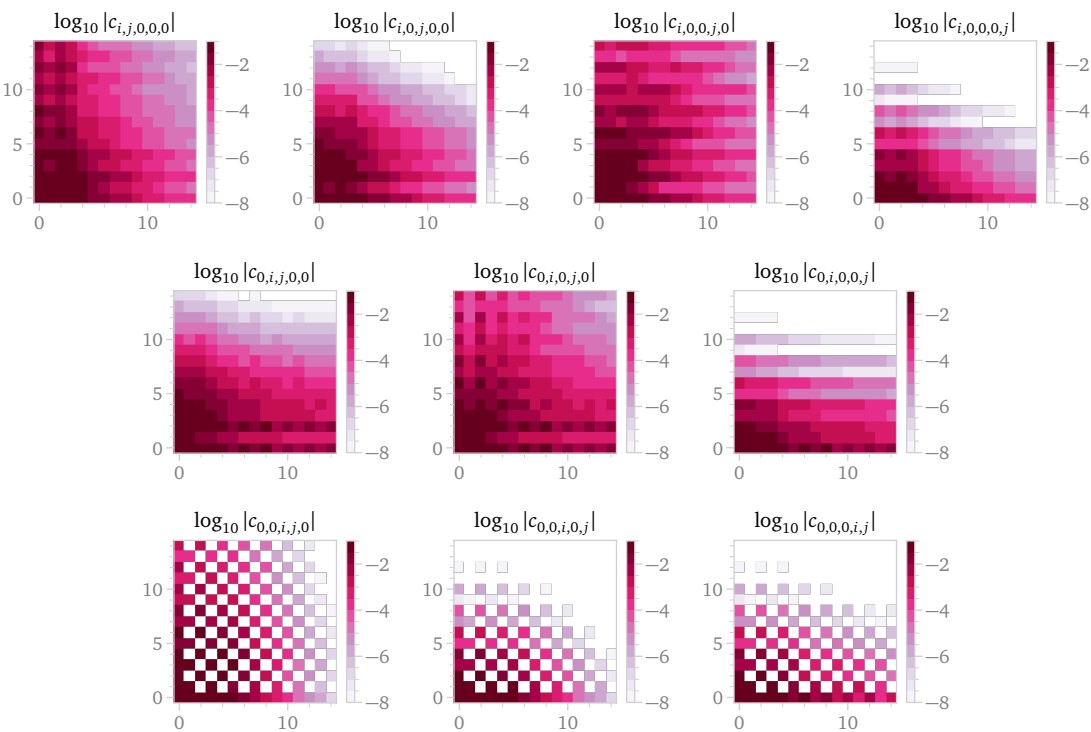

Figure 2: Two-dimensional slices of the 5-dimensional coefficients $c$ corresponding to the decomposition of $f_1$ into a tensor product of Chebyshev polynomials as given in eq. (42). On all plots the horizontal axis is $i$, the vertical is $j$.

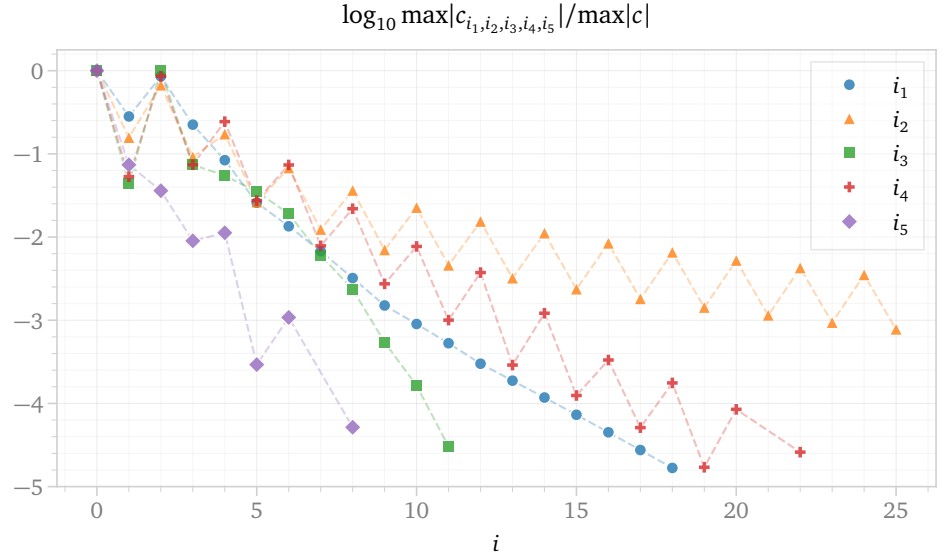

Figure 3: The maximal value of the 5-dimensional coefficients $c$ for $f_1$ along each of the dimensions, corresponding to the decomposition of eq. (42).

Better results would of course be obtained if instead of a fixed ratio, we would choose the best $n_i$ ratio for each value of $n$; however, this can only be done *a posteriori*.

The results of this optimization, together with the non-dimensionally-adaptive version are presented in Figure 4. In this figure we show the approximation errors depending on the number of data points $n$, corresponding to different ways to use polynomial interpolation in practice: different sampling schemes, symmetry handling options, and ways to include weights. The "$a$" method corresponds to interpolating the amplitude $a$ directly, while $f$ corresponds to option 1 from Section 2.4. The methods marked with "half-domain" correspond to option 3 from Section 2.6, while the rest correspond to option 2.

### 3.4 Beyond the full grid

Another way to improve the interpolation is to observe that in Figure 2, even if we optimize the $n_i$ bounds in the cardinal directions, as we did in Section 3.3, we might still be oversampling in the *diagonal* direction: the right upper parts of each slice show significantly smaller coefficients. The reason is simple: diagonals are longer than the edges [52].

A possible solution is to start with the tensor product basis of eq. (42), but use a different set of multi-indices $\vec{i} \equiv \{i_1, \ldots, i_d\}$, such that the upper right corner of Figure 2 is cut out. This will consequentially require choosing a different set of nodes, as we can no longer use the tensor-product construction of eq. (41).

There are many different polynomial bases investigated in the literature (e.g. hyperbolic cross sets [53], lower sets in general [54,55], including adaptively selected ones [56], sparse sets [57]), and many possible choices for the non-tensor-product nodes (e.g. the specific sets like Morrow–Patterson–Xu points [58,59], Padua points [60,61], Chebyshev lattices [62], Lissajous–Chebyshev nodes [63,64], and general constructions for selecting nodes like Fekete points and Leja sequences [65]). Since we cannot hope to faithfully benchmark every possible combination of options in this article, we shall investigate one instance of this general theme as described in [66] using the implementation from [67].

The method consists of constructing the polynomial basis by performing an $L^p$ truncation of the multiindex set in eq. (42), i.e., using all $\vec{i}$ where $\|\vec{i}\|_p \le k$, where $\|\vec{i}\|_p$ is the $L^p$ vector norm, $\left(\sum_i i_i^p\right)^{1/p}$, and choosing the nodes to be the first $n$ points of the tensor-product set of eq. (41) in the Leja order, i.e., chosen one by one greedily, so that the next selected $x_i$ would maximize $\prod_{j<i}(x_i - x_j)$. Results of this method depending on the chosen $p$ are shown in Figure 5.

### 3.5 Effects of noisy data

As a separate important point, we wish to note that the evaluation of $a(\vec{x})$ at any given $\vec{x}$ typically requires more time the higher the precision we demand on it. Thus, it is of practical importance to set a precision target for the evaluations of $a$. Ideally, this should be done such that it does not increase the approximation error. To study how the approximation error is affected by using low-precision values of $a$, we can apply multiplicative noise to it:

$$a_i \to a_i \kappa_i, \quad \kappa \sim \mathcal{N}(1, \sigma^2). \tag{44}$$

The result for different $\sigma$ is shown in Figure 6, where we have intentionally selected very high noise levels to make the effects more visible.

### 3.6 Discussion

Polynomial interpolation is a well understood interpolation method. It achieves exponential convergence rates for analytic functions in one dimension, and is easy to evaluate accurately

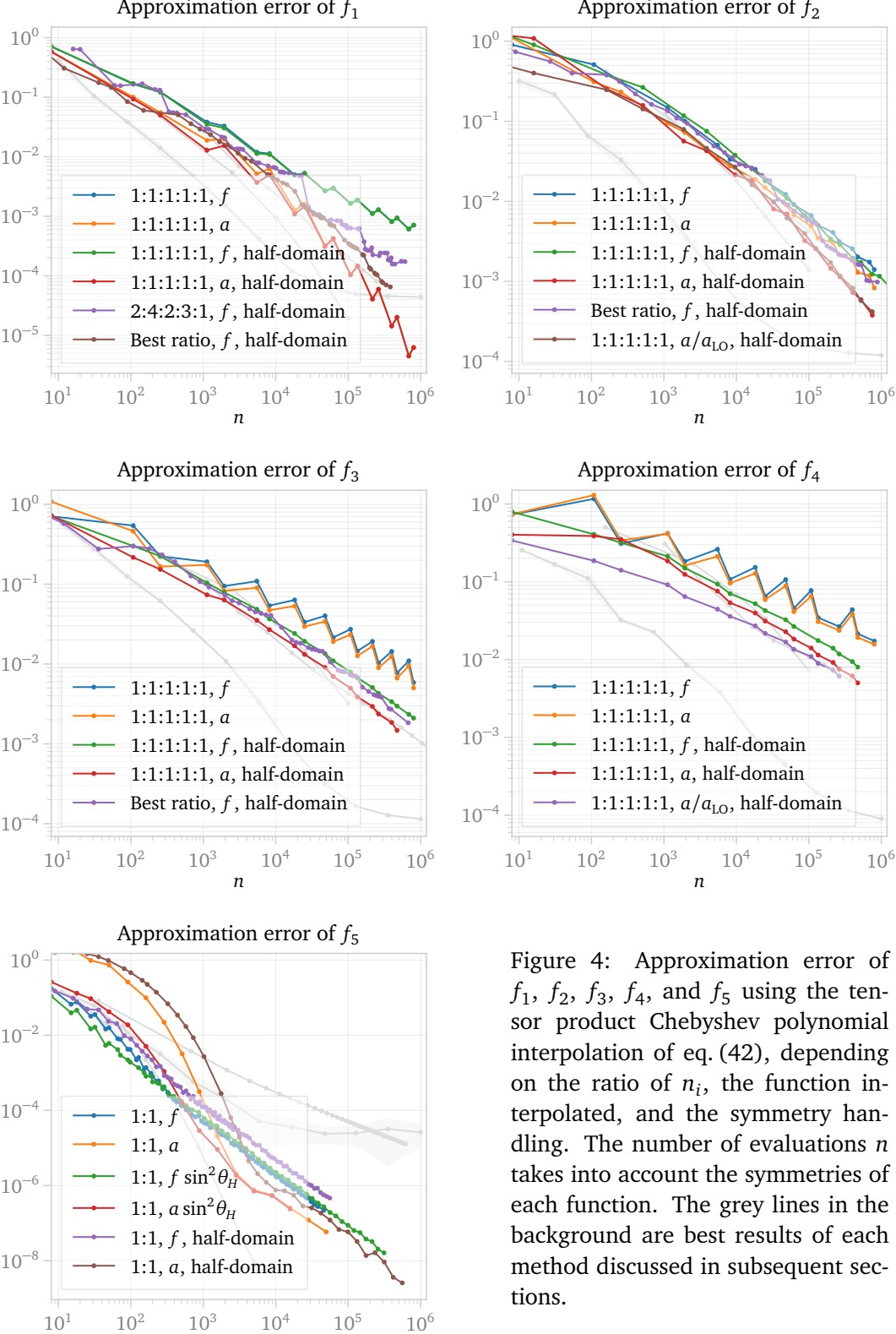

Figure 4: Approximation error of $f_1$, $f_2$, $f_3$, $f_4$, and $f_5$ using the tensor product Chebyshev polynomial interpolation of eq. (42), depending on the ratio of $n_i$, the function interpolated, and the symmetry handling. The number of evaluations $n$ takes into account the symmetries of each function. The grey lines in the background are best results of each method discussed in subsequent sections.

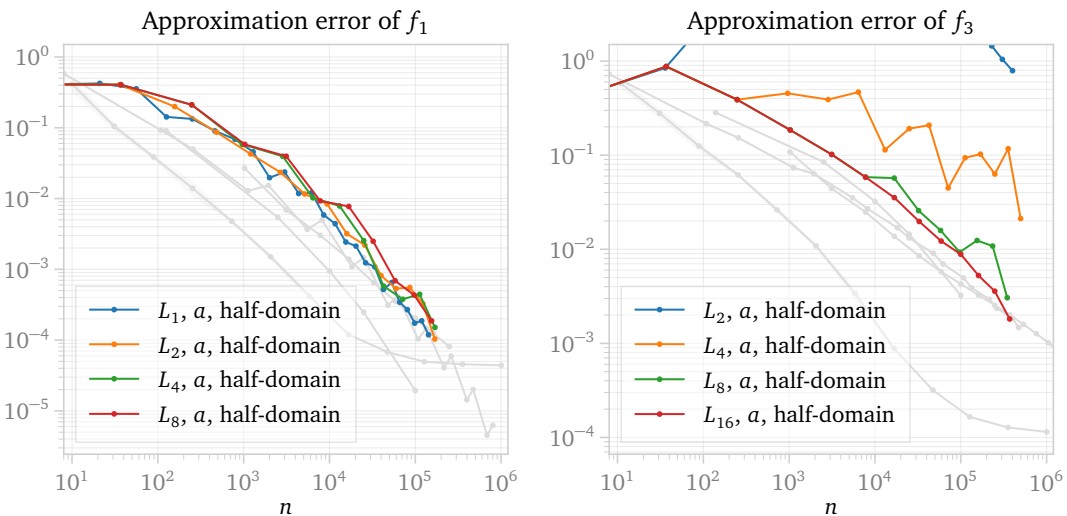

Figure 5: Approximation error (see eq. (6)) of $f_1$ and $f_3$ as a function of the number of training points $n$, using the Leja-ordered non-tensor-product interpolation based on the $L^p$ index set truncation. The "$a$" and "half-domain" labels are the same as in Figure 4.

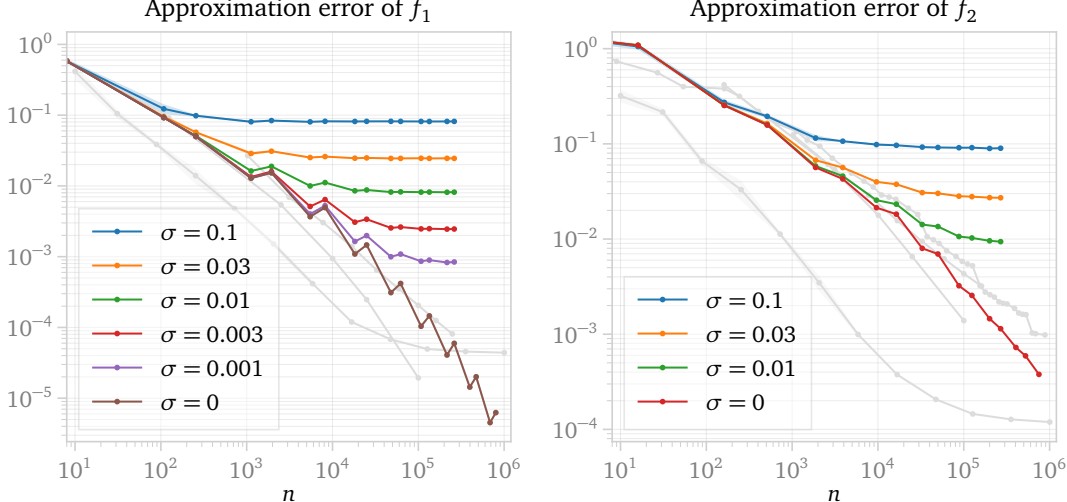

Figure 6: Approximation error (see eq. (6)) of $f_1$ and $f_2$ as a function of the number of training points $n$, if the amplitudes used for evaluation are imprecise, and contain random noise as in eq. (44). The interpolation method corresponds to *1:1:1:1:1, a, half-domain* from Figure 4.

via the barycentric formula. It should be considered a safe and dependable default method.

Its performance in multiple dimensions, however, is held back by the tensor product grid construction, that inherently comes with the curse of dimensionality. It is also sensitive to singularities close to the interpolation space: the closer the singularity, the worse the convergence becomes—this is particularly inconvenient for amplitudes, which are very often not analytic at boundaries. Finally, the method is inflexible: the node sets that do not suffer from the Runge phenomenon are quite rigid, and if an interpolant was constructed with $n$ data points, one can not easily add just a few more—the best one can do is to use a nested node set, as in Section 3, and roughly double $n$. Similarly, if an interpolant was constructed on a subset of the full phase space, there is no good way to smoothly extend it to the rest of the phase space by adding new data.

From the gathered results we can provide insight into the following questions:

- *Should the interpolant be constructed on a or f ?* Including the weight $w$ (i.e., option 1 from Section 2.4) helps a lot for very peaky functions ($f_5$), but seems to slightly hinder others. A possible reason is that $w$ is not analytic at the boundaries, which is a sensitive environment for polynomial interpolation.

- *Does taking a ratio to the leading order help?* Since $f_1$ is the leading-order version of $f_2$, and $f_3$ of $f_4$, it is natural to ask if interpolating $f_2/f_1$ (the so-called *K-factor*) is better than directly $f_2$ or $a_2$. For $f_2$ this appears to make no difference, while for $f_4$ there is a notable improvement at lower precisions.

- *How should the symmetries be included?* Domain reduction (i.e., option 3 from Section 2.6) significantly helps $f_3$ and $f_4$, makes almost no difference for $f_1$ and $f_2$, and slightly hinders $f_5$, for which data duplication (i.e., option 2) is better.

- *Does factoring out the known peaky behaviour from the interpolant help?* Yes, cancelling the $1/\sin^2\theta_H$ factor from the $f_5$ interpolant brings a modest but notable improvement. Subtracting the peak could have been even better though.

- *Does dimensional adaptivity help?* It only makes a notable improvement for $f_1$.

- *Does non-full-grid sampling help?* Not the version of it that we investigated.

- *How precise must the data points be?* The noise magnitude $\sigma$ acts as a lower boundary for $\varepsilon$, so $\sigma$ must be chosen at least as $\sigma \leq \varepsilon_{\text{target}}$. Values of $\sigma \lesssim 1/3\,\varepsilon_{\text{target}}$ seem to allow reaching the desired $\varepsilon$ while being virtually unaffected by noise.

## 4 B-spline interpolation

An alternative way to prevent the Runge phenomenon in polynomial interpolation is to limit the power of the polynomial, and use a basis of splines. A *spline* of degree $p$ is a $p-1$ times continuously differentiable piecewise polynomial. In this section we use *B-splines* (basis splines) [68,69], which is the standard basis choice for spline interpolation.

B-splines have been applied in engineering applications in the context of the finite element method [70] and in graphics as *non-uniform rational B-splines* (NURBS) [71]. B-splines have also been successfully used for interpolating amplitudes in lower-dimensional cases [72,73], which makes this a particularly interesting method for us to compare with. Spline interpolation has also been applied in the context of PDF fits [74].

In this section we define the B-spline basis functions and show how to define multidimensional B-spline interpolants on tensor product grids. We then present the resulting approximation errors from using uniform B-splines and finish with a discussion of the obtained performance.

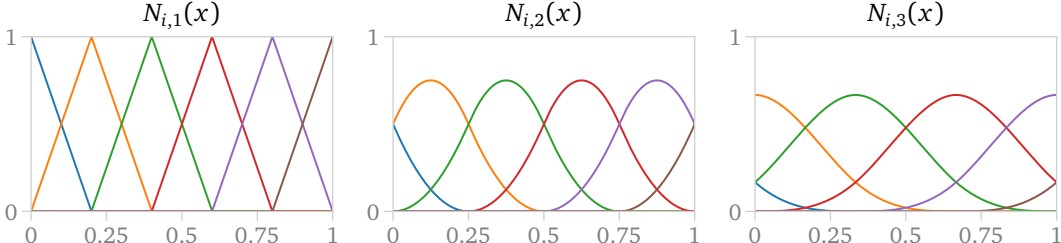

Figure 7: Linear, quadratic and cubic B-spline basis functions from eq. (45) with uniform knot vectors, for different $i$.

## 4.1 B-spline basis functions

A B-spline basis function of degree $p$ and index $i$ is defined recursively through the Cox-de Boor formula [75, 76]:

$$
\begin{aligned}
N_{i,0}(x) &= \begin{cases} 1 & \text{if } t_i \leq x \leq t_{i+1}, \\ 0 & \text{otherwise}, \end{cases} \\
N_{i,p}(x) &= \frac{x - t_i}{t_{i+p} - t_i} N_{i,p-1}(x) + \frac{t_{i+p+1} - x}{t_{i+p+1} - t_{i+1}} N_{i+1,p-1}(x),
\end{aligned}
\tag{45}
$$

where $t_i$ are the coordinates of *knots*—the locations at which the piecewise functions meet.

Knots define a *knot vector* $\vec{t}$, which is a sequence of $m + 1$ non-decreasing numbers $(t_0, \ldots, t_m)$. The number of knots is related to the degree through $m = n + p + 1$, where $n + 1$ is the number of basis functions [77]. In an interpolation context, the first and last $p + 1$ knots should be placed outside or at the boundary of the interpolation domain, such that the lower and upper boundaries correspond to $t_p$ and $t_{m-p}$. Under these conditions the *local Marsden identity* is fulfilled, which ensures that the B-spline basis represents polynomials exactly on the domain $(t_p, t_{m-p})$ [78, 79]. This ensures that the B-spline has approximation power across the entire phase space. The most straightforward choice of knot vector is therefore the uniform construction, with $p$ auxiliary knots placed outside the interpolation domain. The basis functions resulting from such uniform knot vectors for $p = 1, 2, 3$ are shown in Figure 7.

Another common choice is the *not-a-knot* construction, where knots coincide with data points, except at $p - 1$ points, which are omitted. Since in the even degree case this results in an odd number of not-a-knots, this knot vector is better defined for odd degree B-splines. Note that the linear case simplifies to the uniform construction since there are 0 not-a-knots in this case.

From eq. (45) it can be seen that the basis functions are non-negative, form a partition of unity and have local support. Moreover, thanks to the recursive nature of eq. (45), there are efficient algorithms that make B-splines fast to evaluate compared to other spline functions [77]. This is especially true for B-splines with uniform knot vectors, since the denominators in eq. (45) are constant. In the not-a-knot case the knot distances are not all equal, but can be precomputed in advance.

## 4.2 B-spline interpolants

A one-dimensional B-spline interpolant is a linear combination of $n + 1$ basis functions

$$
b(x) = \sum_{i=0}^{n} N_{i,p}(x) \cdot c_i,
\tag{46}
$$

where the $c_i$ are interpolation coefficients sometimes referred to as *control points*. The most straightforward extension to the $d$-dimensional case is through a tensor product construction

$$B(\vec{x}) = \prod_{j=1}^{d} b_j(\vec{x}_j). \tag{47}$$

To fully constrain a B-spline with $n+1$ basis functions in each direction, $(n+1)^d$ interpolation nodes are required. The interpolation nodes can partially be selected freely but need to satisfy the Schoenberg–Whitney conditions, which are degree dependent and state that for each knot $t_i$ there must be at least one data point $x$ such that $t_i < x < t_{i+p+1}$. Moreover, the properties of the basis functions imply that a B-spline interpolant is numerically stable [80].

### 4.3 Discussion

In Figure 8 the performance of B-spline interpolation is compared to the other methods described in this paper. For all test functions a significant performance gain over linear splines is obtained by using basis functions of at least quadratic degree. For $f_1$ and $f_2$ cubic splines perform slightly better than quadratic ones; for $f_3$, $f_4$, and $f_5$ quadratic splines are the best.

The effect of interpolating either the amplitude ("$a$") or the test function ("$f$") is shown for $f_1$, $f_3$ and $f_5$. For $f_1$ and $f_3$ it is better to directly interpolate the amplitude. For $f_5$ including the weight helps in the low training-data regime, but becomes worse for high amount of training data.

B-splines suffer from the same flexibility issues as polynomials, as discussed in Section 3.6, because the extension of the B-splines to the multivariate case is based on the same tensor product construction. It is difficult to predict exactly how many points are required for a certain precision target, so the required number of evaluations likely overshoots the minimal number of evaluations. Therefore, the results in Figure 8 are optimistic from a practical point of view. We describe in Section 5 how B-splines can be combined with an adaptive approach in the context of sparse grids, which removes this problem.

Additionally, B-splines require function evaluations at the boundaries, which is also pointed out in Section 3.6 as a drawback for amplitude interpolation. For example, it is observed in [37] that the evaluation time diverges close to certain boundaries. For such cases, the boundary of the interpolation space can be shifted to a point where the evaluation time is reasonable, meaning that the approximation would not cover some parts of the phase space. Alternatively, methods that incorporate extrapolation, such as the modified bases on adaptive sparse grids from Section 5, can be used to alleviate this problem.

Where B-splines shine is in their evaluation speed: a B-spline interpolant can be evaluated in small constant time, whereas a polynomial interpolant needs time proportional to the number of data points. For this reason, B-splines can speed up the evaluation of more precise approximation methods by first constructing the approximation using those methods, and then approximating that approximation using B-splines. This is possible because the data points for the second approximation will be obtained by evaluating the first approximation, which should be much faster than evaluating the target function.

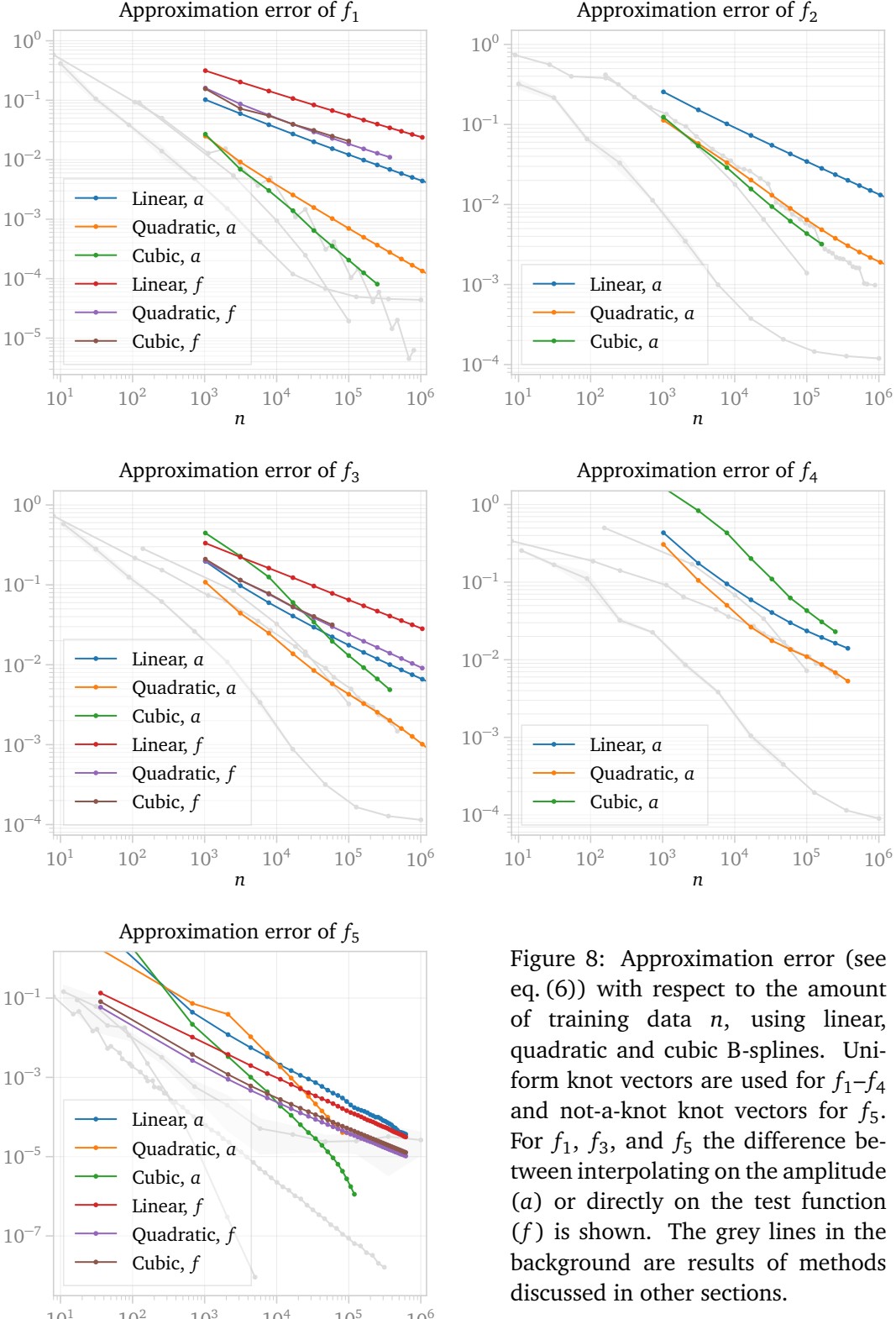

Figure 8: Approximation error (see eq. (6)) with respect to the amount of training data $n$, using linear, quadratic and cubic B-splines. Uniform knot vectors are used for $f_1$–$f_4$ and not-a-knot knot vectors for $f_5$. For $f_1$, $f_3$, and $f_5$ the difference between interpolating on the amplitude ($a$) or directly on the test function ($f$) is shown. The grey lines in the background are results of methods discussed in other sections.

# 5 Sparse grids

The methods described in the previous sections are based on the tensor product construction of eq. (41), which results in a *full grid*. A *sparse grid* construction aims to omit points from the full grid that do not significantly contribute to the interpolant, and in this way alleviate the curse of dimensionality. Such constructions were first described by Smolyak [81], then rediscovered by Zenger [82] and Griebel [83], and have since found use in a wide variety of applications [84–91].

In this section we first introduce sparse grids built on a hierarchical linear basis. Next, we describe how to incorporate spatial adaptivity and upgrade the basis to higher degree polynomials. Finally, we study the impact these constructions have on the approximation quality.

## 5.1 Classical sparse grids

Sparse grids can be constructed in two main ways, either with the *combination technique* using a linear combination of full grids [55], or through a hierarchical decomposition of the approximation space [86]. In this section we use the latter approach, since it makes it straightforward to incorporate spatial adaptivity.

First, let us restrict to functions that vanish at the boundaries. In this case, a one-dimensional basis can be constructed with rescaled "hat" functions that are centred around the grid points $x_{l,i} = i \cdot 2^{-l}$, $i \in \{0, 1, ..., 2^l\}$:

$$\phi_{l,i}(x) \equiv \phi\left(\frac{x - i \cdot 2^{-l}}{2^{-l}}\right), \qquad \phi(x) \equiv \max(1 - |x|, 0), \tag{48}$$

where $l$ is the *grid level*. Since these basis functions have local support, it is possible to define *hierarchical subspaces* $W_l$ through the *hierarchical index* sets $I_l$ by

$$W_l \equiv \text{span}\{\phi_{l,i} : i \in I_l\}, \qquad I_l \equiv \{i \in \mathbb{N} : 1 \le i \le 2^l - 1, i \text{ is odd}\}. \tag{49}$$

The function space of one-dimensional interpolants is then defined as the direct sum of all subspaces up to a maximum grid level $k$

$$V_k \equiv \bigoplus_{l \le k} W_l. \tag{50}$$

The extension to the multivariate case is via a tensor product construction

$$\Phi_{\vec{l},\vec{i}}(\vec{x}) \equiv \prod_{j=1}^{d} \phi_{l_j,i_j}(x_j), \tag{51}$$

with multi-indices $\vec{i} = (i_1, ..., i_d)$, $\vec{l} = (l_1, ..., l_d)$ and $d$-dimensional grid points $\vec{x}_{\vec{i},\vec{l}} = (x_{l_1,i_1}, ..., x_{l_d,i_d})$. Similarly to the one-dimensional case, the hierarchical subspaces are defined to be

$$W_{\vec{l}} \equiv \text{span}\{\Phi_{\vec{l},\vec{i}} : \vec{i} \in I_{\vec{l}}\}, \qquad I_{\vec{l}} \equiv \{\vec{i} : 1 \le i_t \le 2^{l_t} - 1, i_t \text{ is odd}, 1 \le t \le d\}. \tag{52}$$

A multidimensional full-grid function space $V_k^F$ can now be naively constructed with the direct sum

$$V_k^F \equiv \bigoplus_{|\vec{l}|_\infty \le k} W_{\vec{l}}, \tag{53}$$

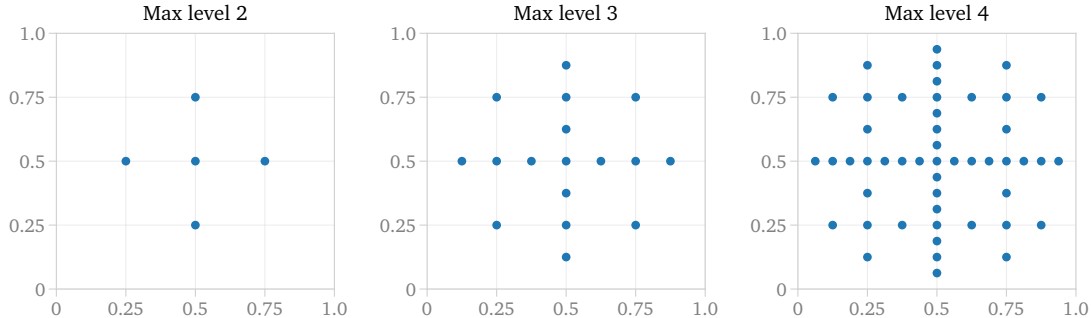

Figure 9: Sparse grid structure in two dimensions for $k = 2, 3, 4$.

| Grid type | $_d \backslash ^k$ | 2 | 4 | 6 | 8 | 10 |
|---|---|---|---|---|---|---|
| **Full** | $d = 2$ | 25 | 289 | 4225 | $6.6 \cdot 10^4$ | $1.1 \cdot 10^6$ |
| | $d = 5$ | 3125 | $1.4 \cdot 10^6$ | $1.2 \cdot 10^9$ | $1.1 \cdot 10^{12}$ | $1.1 \cdot 10^{15}$ |
| **Sparse** | $d = 2$ | 5 | 49 | 321 | 4097 | $9.2 \cdot 10^3$ |
| | $d = 5$ | 11 | 315 | 5503 | $6.1 \cdot 10^4$ | $5.5 \cdot 10^5$ |

Table 1: Number of grid points from the full and sparse constructions, for dimensions 2 and 5.

where $|\vec{l}|_\infty = \max(l_1, \ldots, l_d)$. The mechanism of the sparse grid is to instead limit the selection of subspaces according to

$$V_k^S \equiv \bigoplus_{|\vec{l}|_1 \leq k+d-1} W_{\vec{l}}, \tag{54}$$

where $|\vec{l}|_1 = l_1 + \cdots + l_d$. This avoids including basis functions that are highly refined in all directions simultaneously, which is what causes the number of grid points to explode in high dimensional cases.

Figure 9 shows nodes of the resulting two-dimensional sparse grids for $k = 2, 3, 4$. Table 1 shows the difference in the number of grid points between sparse and full constructions, for dimensions 2 and 5. Omitting grid points can never increase the approximation quality, but the aim is to achieve a high approximation quality while omitting most points. For sufficiently smooth functions, it is known that the asymptotic accuracy of a full grid interpolant scales with the mesh-width $h_k = 2^{-k}$ as $\mathcal{O}(h_k^2)$, while for a sparse grid it decreases only by a logarithmic factor to $\mathcal{O}(h_k^2 (\log h_k^{-1})^{d-1})$ (see [86]).

## 5.2   Boundary treatment

Since the basis functions of eq. (48) do not have support at the grid boundaries, we can so far only handle target functions that vanish at the boundary. There are two main ways to incorporate boundary support into sparse grids. One possibility is to add boundary points to the grid [86], but for even moderately high dimension this causes the number of points to increase significantly, and defeats the main purpose of the sparse construction. The other option is to use so-called *modified* basis functions [86] that linearly extrapolate from the outermost grid

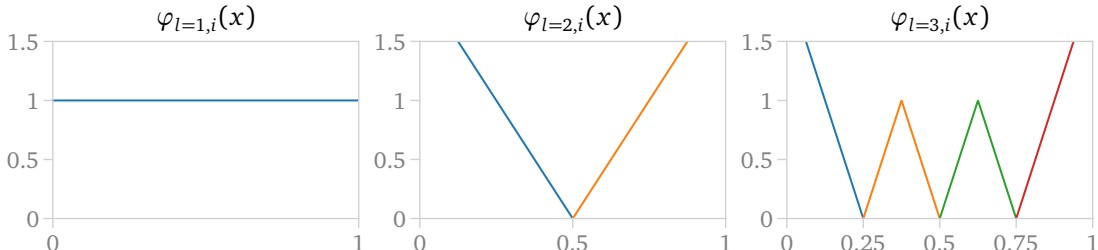

Figure 10: Modified linear hat-basis for sparse grid levels $l = 1, 2, 3$ and different $i$.

points:

$$
\varphi_{l,i}(x) \equiv
\begin{cases}
1 & \text{if } l = 1 \land i = 1, \\[4pt]
\begin{cases} 2 - 2^l \cdot x & \text{if } [0, \frac{1}{2^{l-1}}], \\ 0 & \text{otherwise.} \end{cases} & \text{if } l > 1 \land i = 1, \\[14pt]
\begin{cases} 2^l \cdot x + 1 - i & \text{if } x \in [1 - \frac{1}{2^{l-1}}, 1], \\ 0 & \text{otherwise.} \end{cases} & \text{if } l > 1 \land i = 2^l - 1, \\[14pt]
\phi_{l,i}(x) & \text{otherwise.}
\end{cases}
\tag{55}
$$

Figure 10 shows the modified linear basis for $k = 1, 2, 3$. It is apparent that sparse grids are best suited for functions whose boundary behaviour is less important than that in the bulk to the overall structure. The interpolant is constructed as a linear combination of basis functions in $V_k^S$ according to

$$
u(\vec{x}) = \sum_{|\vec{l}|_1 \leq k + d - 1} \sum_{\vec{i} \in I_{\vec{l}}} \alpha_{\vec{l}, \vec{i}} \cdot \Phi_{\vec{l}, \vec{i}}(\vec{x}),
\tag{56}
$$

where the interpolation coefficients $\alpha_{\vec{l}, \vec{i}}$ are referred to as *hierarchical surpluses* since at each level they correct the interpolant from the previous level to the target function. They are thus also a measure of the absolute error at each level in each direction, which makes the interpolant $u(\vec{x})$ well suited for local adaptivity. The basis functions $\Phi_{\vec{l}, \vec{i}}(\vec{x})$ are constructed from either eq. (48) or eq. (55) depending on if the target function vanishes at the boundary or not. The process of determining the coefficients $\alpha_{\vec{l}, \vec{i}}$ is known as *hierarchization*. This can in principle be done in the same way as for any interpolation method: construct and solve a linear system of equations where the interpolant is demanded to reproduce the target function at each interpolation node. However, a more efficient way is available for bases satisfying the *fundamental* property:

$$
\varphi_{l,i}(2^{-l} i) = 1 \quad \text{and} \quad \varphi_{l,i}(2^{-l}(i-1)) = \varphi_{l,i}(2^{-l}(i+1)) = 0,
\tag{57}
$$

In this case the coefficients can be calculated on the fly: the grid is initialized by normalizing the first coefficient to the central value $\alpha_{\vec{1}, \vec{1}} = f(0.5, \ldots, 0.5)$; points $\vec{x}_{\vec{l}, \vec{i}}$ are then added one at a time, and the coefficients are defined as the corrections

$$
\alpha_{\vec{l}, \vec{i}} = f(\vec{x}_{\vec{l}, \vec{i}}) - \tilde{f}(\vec{x}_{\vec{l}, \vec{i}}),
\tag{58}
$$

where $\tilde{f}(\vec{x}_{\vec{l}, \vec{i}})$ is the current approximation with the already added points and $f(\vec{x}_{\vec{l}, \vec{i}})$ is the true function value.

## 5.3  Spatially adaptive sparse grids

While the classical sparse grid construction uses significantly fewer points than a full grid, it is completely blind to how the target function varies across the parameter space. In many

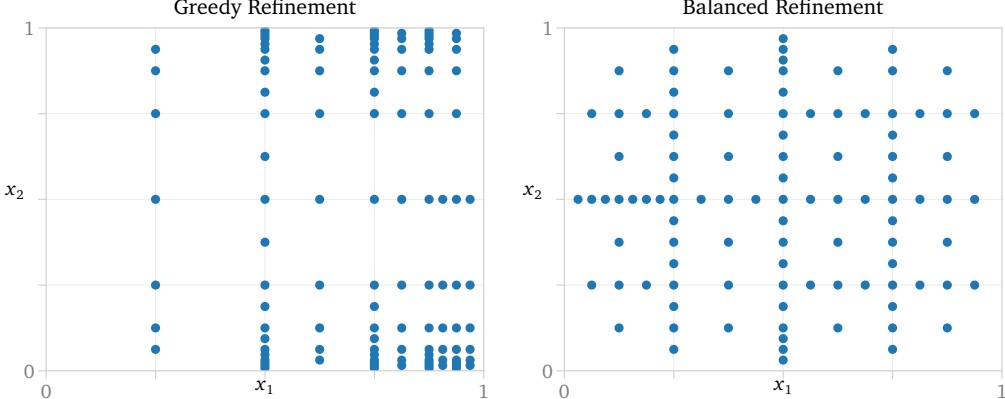

Figure 11: Point distribution from spatial adaptivity with greedy and balanced refinement for the two-dimensional test function $f_5$.

high dimensional problems, it is typical that some regions are more important than others. This information is exploited by spatially adaptive sparse grids, where the grid points that contribute most to the interpolant are *refined* first (see [86,92]). Refining a grid point means adding all its neighbouring points at one level lower to the grid.

Determining which points are more important relies on some heuristic criterion. A common strategy is to use the *surplus criterion*, where the point with the largest hierarchical surplus is refined first. This is the *greedy strategy*. Its disadvantage is that it might get stuck refining small regions of the parameter space. To avoid this, [86] proposes to weigh the surplus criterion by the volume of the support of the corresponding basis function, $2^{-|\vec{l}|_1}$, resulting in the *balanced strategy*.

Figure 11 shows the difference between greedy and balanced refinement on a two-dimensional grid, for the test function $f_5$. We find that greedy refinement might perform slightly better for our error metric if the structure of the target function is simple enough, while balanced refinement is significantly more reliable for more complicated target functions.

## 5.4 Higher degree basis functions

The one-dimensional basis of rescaled hat functions in eq. (48) is straightforward to implement and avoids the Runge phenomenon due to its piecewise nature. The drawback is that the linear approximations come with low interpolation performance, as we have previously observed in Section 4.

There are many ways to make the extension to higher polynomial degrees. Here we differentiate between two types of basis functions: those that fulfil the fundamental property eq. (57) and those that do not.

Fundamental basis functions allow for fast hierarchization and make it easy to construct efficient evaluation algorithms. In addition to the linear basis from Section 5.1, we test the more general polynomial piecewise functions (C0-elements) [86,93], as well as fundamental B-splines [89]. For these benchmarks we make use of the sparse grid toolbox SG++ [86], where both of these basis functions are already implemented. Beyond fundamental basis functions we also investigate extended not-a-knot B-splines as described in [90,94].

### C0 elements

Higher degree polynomials can be defined on the hierarchical structure by using grid points on upper levels as the polynomial nodes [86,93]. This implies the maximum degree $p$ is bounded

| $p$ | $e_{i,j=0}, \quad i = 1, \ldots, p+1$ | $e_{i,j=2^l}, \quad i = 2^l - 1 - p, \ldots, 2^l - 1$ |
|---|---|---|
| 1 | $[2, -1]$ | $[-1, 2]$ |
| 3 | $[5, -10, 10, -4]$ | $[-4, 10, -10, 5]$ |
| 5 | $\begin{cases} [8, -28, 42, -35, 20, -6] & \text{if } l = 3, \\ [8, -28, 56, -70, 56, -21] & \text{if } l > 3. \end{cases}$ | $\begin{cases} [-6, 20, -35, 42, -28, 8] & \text{if } l = 3, \\ [-21, 56, -70, 56, -28, 8] & \text{if } l > 3. \end{cases}$ |

Table 2: B-spline extension coefficients for the upper and lower boundaries, taken from [94].

by the maximum grid level. For sparse grids without boundaries the relation is $p \leq l - 1$. This can be seen already for the linear case in Figure 10, since on levels 1 and 2 the basis functions are constant and linear respectively. The drawback of this extension is that despite the higher degree, the smoothness is not increased and the interpolant will only be continuous. For this reason, these basis functions are usually referred to as C0 elements.

**Fundamental B-splines**

The fundamental B-spline basis is constructed with the aim of fulfilling the fundamental property of eq. (57), while preserving useful properties of B-splines such as smoothness. The construction is introduced in [89], and it works by applying a translation-invariant fundamental transformation to the hierarchical B-spline basis. Besides fulfilling the fundamental property, this transformation preserves the translational invariance of B-splines which improves performance during evaluation. The modified basis that extrapolates toward the boundaries is defined similarly to the linear case. We refer to [89] for more details on the derivation. For both this and the C0 elements, the implementations in SG++ are used for the benchmarks.

**Extended not-a-knot B-splines**

In this section we summarize the main equations and statements on extended not-a-knot B-splines presented in [90, 94]. The extension mechanism ensures that polynomials are interpolated exactly, which in many cases increases the quality of interpolation. The basis consists of extended not-a-knot B-splines on lower levels, and Lagrange polynomials on upper levels where there are too few points for the B-spline to be defined. A basis function of odd degree $p$, level $l$ and index $i$ is defined as

$$b_{l,i}^{p,\text{ext.}}(x) = \begin{cases} b_{l,i}^p + \sum_{j \in J_l(i)} e_{i,j} \, b_{l,j}^p & \text{when } l \geq \Lambda^{\text{ext.}}, \\ L_{l,i}(x) & \text{when } l < \Lambda^{\text{ext.}}, \end{cases} \tag{59}$$

where $\Lambda^{\text{ext.}} = \lceil \log_2(p+2) \rceil$, and $e_{i,j}$ are extension coefficients that only depend on the degree. For $p = 1, 3, 5$, these coefficients are listed in Table 2. The $b_{l,i}^p$ are not-a-knot B-splines (see Section 4) and $L_{l,i}(x)$ are regular Lagrange polynomials defined with a uniform knot vector $t = (0, 2^{-l}, \ldots, 1)$ as

$$L_{l,i}(x) = \prod_{\substack{1 \leq m \leq 2^l - 1, \\ m \neq i}} \frac{x - t_m}{t_i - t_m}, \quad i = 0, \ldots, 2^l. \tag{60}$$

The index set $J_l(i)$ defines the extension of the B-spline. It determines to which interior basis functions the boundary basis functions are added. An interior basis function of index $i$ includes the boundary basis function if $i$ is among the first or last $p + 1$ interior indices. Formally the index set is defined as $J_l(i) = \{j \in J_l \mid i \in I_l(j)\}$, where $J_l = \{0, 2^l\}$ and

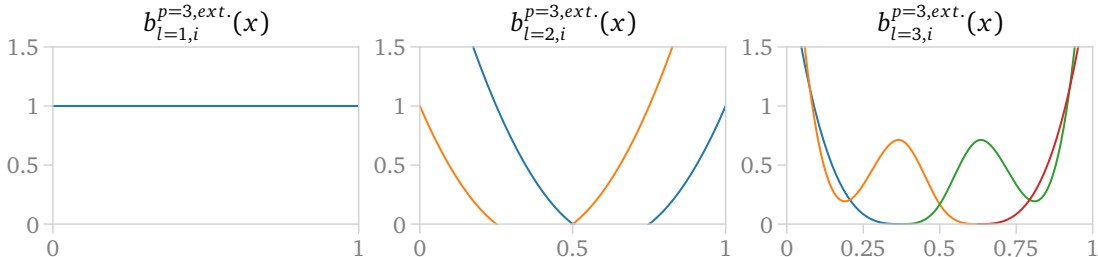

Figure 12: Extended not-a-knot cubic B-spline basis functions at sparse grid levels $l = 1, 2, 3$ and different $i$. At level 1 there are not enough points to define the B-spline and a linear Lagrange polynomial is used instead.

$I_l(0) = \{1, \ldots, p+1\}$, $I_l(2^l) = \{2^l - 1 - p, \ldots, 2^l - 1\}$ (see [94]). The basis functions described by eq. (59) are shown in Figure 12 for the cubic case. The linear case simplifies to the linear hat-basis described in Section 5.1.

## 5.5 Discussion

In Figure 13 we compare the approximation error from spatially adaptive sparse grids constructed with balanced refinement against the other methods described in this paper. A comparison between different refinement criteria is given in Figure 14 for test functions $f_1$, $f_3$, and $f_5$, using the modified linear basis. In some cases the greedy construction is slightly better, but for example the results for $f_3$ demonstrate the danger of this refinement strategy. The balanced construction results in a much smaller approximation error, which hints at the greedy algorithm getting stuck refining local structures. In practice, it is difficult to predict when refining greedily is better, and in such cases the advantage seems to not be very significant. Moreover, greedy refinement is sensitive to noise in the training data due to random enhancement of hierarchical surpluses. Finally, balanced refinement is more robust against noise since it suppresses the effect of local deviations by construction. For these reasons, balanced refinement is the more reliable approach for our studies.

We also try to weigh the refinement criterion by the phase-space weight $w$ from eq. (29). For $f_1$ and $f_3$ this results in a more uniform target function, and the resulting sparse grid becomes similar to the balanced construction without weighting. We therefore see only minor differences in the results from balanced and weighted refinement.

For other basis choices we see significant improvements for all test functions when increasing the degree with the piecewise polynomial and fundamental B-spline bases. With the extended not-a-knot B-splines we see an even better improvement for $f_5$, but for $f_1$–$f_4$ the performance is at best similar to the linear case. In particular for $f_3$ and $f_4$, the cubic and quintic cases converge very poorly. This result is unexpected since these bases have been applied to high-dimensional test functions with good results in [94]. As a cross-check, we have reproduced those benchmarks with our implementation and our results with SG++.

For the loop amplitudes $f_2$ and $f_4$ we also try to flatten the target function by interpolating their ratios to the corresponding leading-order amplitudes ($f_1$ and $f_3$). Figure 13 shows the results for the linear, polynomial and fundamental spline bases. We see a significant improvement in the low-data regime, but the scaling is worse compared to interpolating the amplitude directly. Additionally, the higher degree basis functions appear to be more robust against the worse scaling.

The improvement of the spatially adaptive sparse grids over the non-adaptive full grid methods at low amounts of training data is modest. However, we observe a better scaling

overall for each test function, yielding a significant performance boost at high data regimes.

Taking interpolation performance aside, a major advantage of spatially adaptive sparse grids is their flexibility. Since it is difficult to predict how much training data is required to reach a certain precision target for an unknown function, it is likely that training data needs to be added iteratively. As is discussed in Section 3 and Section 4, non-adaptive methods are very limited in this regard. A spatially adaptive sparse grid on the other hand is able to add one point at a time, making it possible to validate during construction and in principle stop at exactly the required number of points. If a non-adaptive method is used to reconstruct an unknown function, we are likely to overshoot the required number of points, making the effective number of function evaluations higher than what is represented by these results. The results for the non-adaptive methods at a given training size are therefore the best case scenario, while the sparse grid results are closer to what one obtains in practice.

Another benefit is the evaluation performance: for sparse grids it scales linearly with the maximal depth, which is logarithmic in the number of data points.

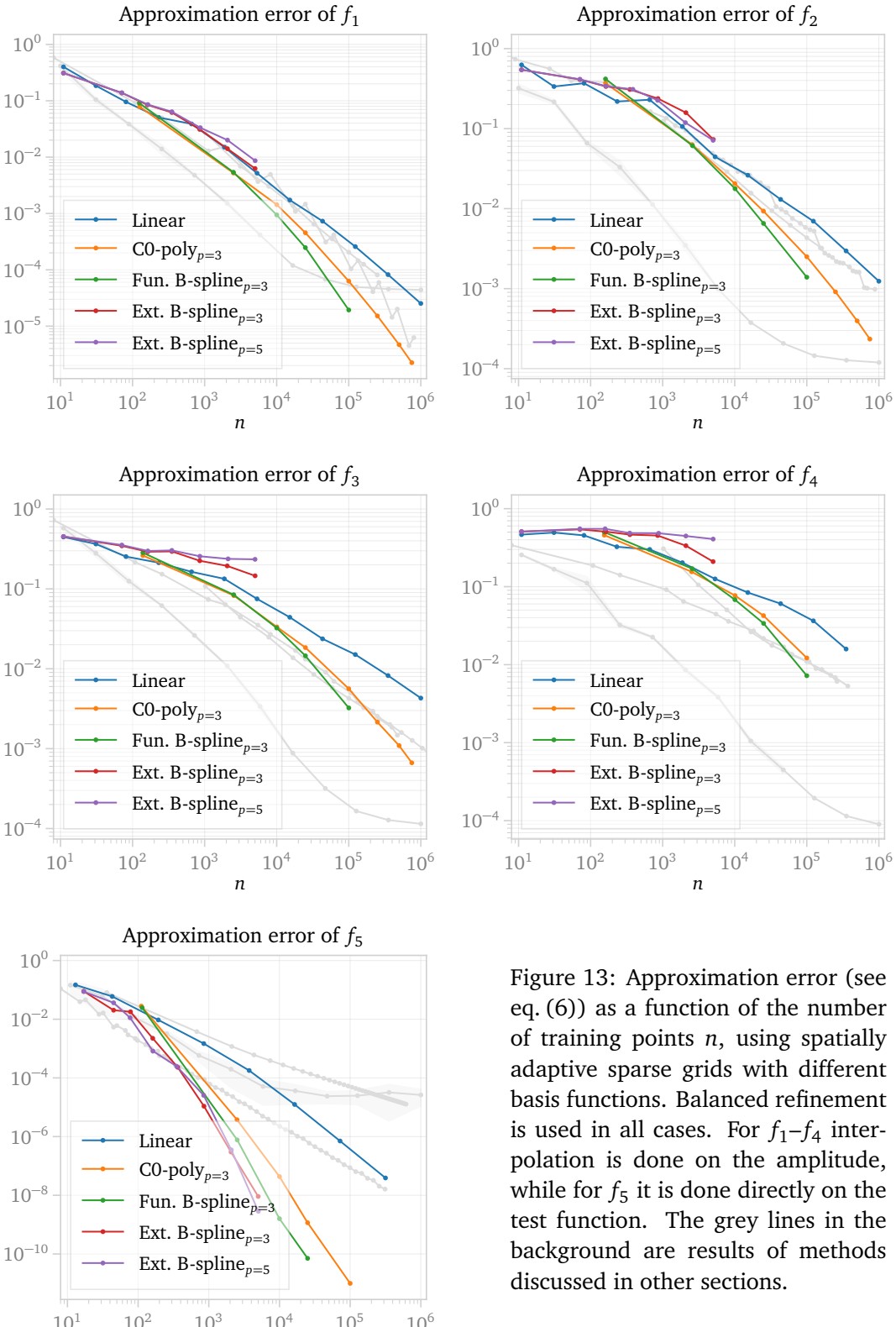

Figure 13: Approximation error (see eq. (6)) as a function of the number of training points $n$, using spatially adaptive sparse grids with different basis functions. Balanced refinement is used in all cases. For $f_1$–$f_4$ interpolation is done on the amplitude, while for $f_5$ it is done directly on the test function. The grey lines in the background are results of methods discussed in other sections.

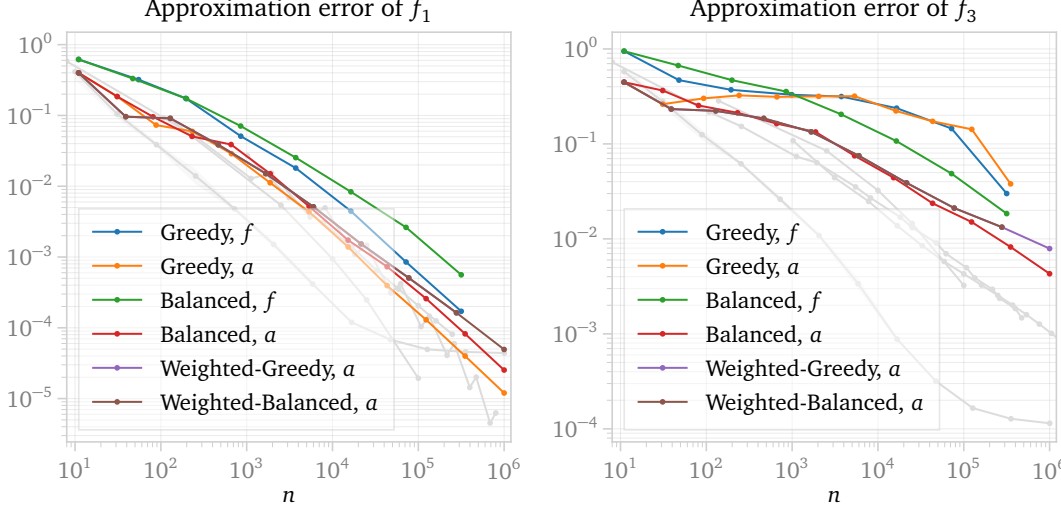

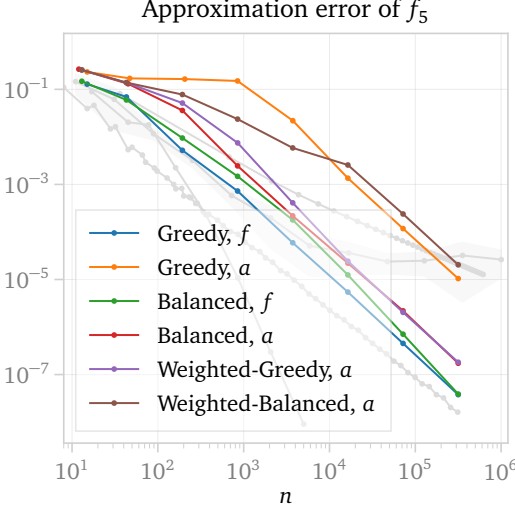

Figure 14: Effect of refinement criterion and choice of target function on the approximation error (see eq. (6)) as a function of the number of training points $n$, for $f_1$, $f_3$, and $f_5$. The difference between interpolating on the amplitude ($a$) as well as on the test function ($f$) is shown. In addition, we try interpolating on the amplitude while suppressing the refinement condition with the phase-space weight. For the comparison in this figure the linear basis is used.

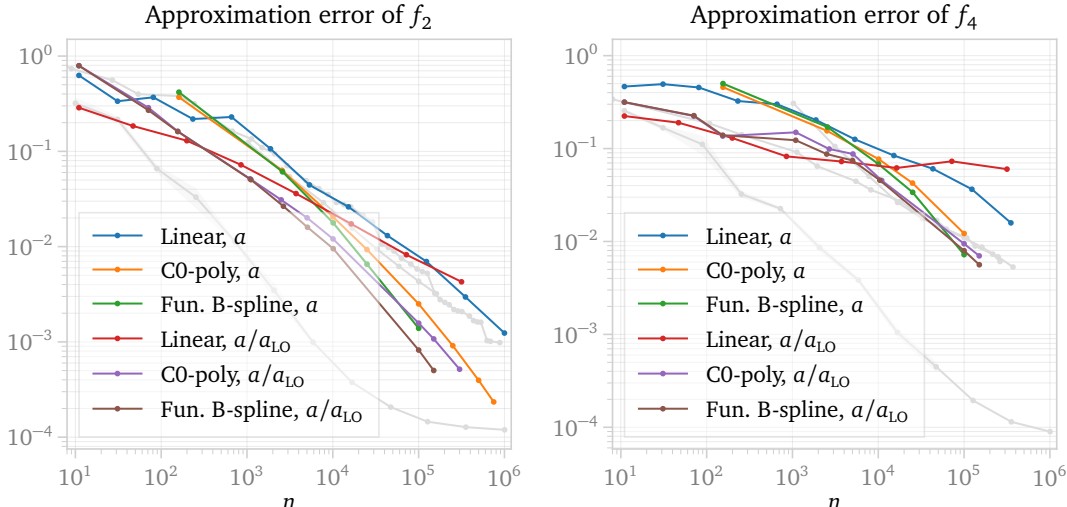

Figure 15: Approximation error (see eq. (6)) as a function of the number of training points $n$, using sparse grids with balanced refinement targeting amplitudes $a$ and the ratios $a/a_{\mathrm{LO}}$ respectively. The polynomial and B-spline basis functions are all of cubic degree in this figure. The grey lines in the background are results of methods discussed in other sections.

# 6 Machine Learning Techniques

Machine learning techniques are a very promising tool for amplitude interpolation [95–100]. These techniques leverage the power of neural networks, approximator functions that are structured as a nested sequence of operations dependent on learnable parameters. Neural networks can be optimized to approximate any function of a set of inputs to arbitrary precision—given sufficient time and data. This is accomplished by stochastically minimizing the *loss function*, which characterizes how close the neural network results are to the desired ones. The optimization of neural network parameters is performed by evaluating the loss function on data subsets or batches and performing gradient descent to minimize the objective in an iterative way. In our case, neural network trainings are framed as regression tasks, where the output of the network is trained to match a target given an input.

The main advantage of this approach for the interpolation problem is its versatility. Neural networks by default introduce a minimal bias in the interpolants they represent, so they can easily adapt to a wide variety of target distributions. Additionally, networks are not limited to a single training or dataset for their optimization. This enables their usage for adaptive interpolation, since after an initial training their weakest predictions can be used as a guideline for optimization through subsequent retrainings.

However, the prediction quality of neural amplitude surrogates declines substantially as particle multiplicity grows [97]. Their performance also decreases with the inclusion of beyond leading-order contributions; exploiting the known structure of infrared divergent QCD radiation can mitigate issues related to peaked integrands [98]. In our case, we tackle the increased amplitude complexity by including prior knowledge about physical symmetries into the learning task. We focus on the Lorentz symmetry and work with architectures that are aware of the Lorentz invariance of the amplitudes. This feature is also present in all other interpolation methods discussed in this paper, but there are several ways to incorporate it into neural networks. We explore two ways of enforcing Lorentz invariance. On the one hand, we train a *multilayer perceptron* (MLP) [101, 102] by feeding it only Lorentz invari-

ant inputs. On the other hand, we use the *Lorentz-Equivariant Geometric Algebra Transformer* (L-GATr) [99, 100], a neural network whose operations are constrained to be equivariant (or covariant) with respect to Lorentz transformations.

To approximate our test functions, we train the networks on $a_n$ as functions of the phase-space points using the test function from eq. (30) as a loss function. We generate two sets of training data points, one sampled uniformly over the phase space, using a low-discrepancy Sobol sequence, and another using the leading-order-unweighted sampling described in eq. (30). Training on the second set corresponds to option 1 from Section 2.4. We also generate unweighted samples for the testing data points. Additionally, we also build extra datasets from unweighted samples for *validation*, a routine checkup that is performed regularly during training to prevent overfitting and select the best performing model constructed during training. The validation set size is always 10% of that of the training dataset until it reaches a size of 4000 points; after that it remains constant for larger training sets.

For $f_1$ and $f_3$ our networks output $a$ directly, while for $f_2$ and $f_4$ they output $a/a_{\text{LO}}$. Due to a larger range in the amplitude distribution, for $f_5$ we preprocess the amplitude targets by using logarithmic *standardization*:

$$\hat{a}_5 = \frac{\log(a_5) - \overline{\log(a_5)}}{\sigma_{\log(a_5)}}, \tag{61}$$

where $\overline{\log(y_i)}$ and $\sigma_{\log(y_i)}$ are the mean and standard deviation of the amplitude logarithm distributions over the whole dataset. In this case, the preprocessed amplitude $\hat{a}_5$ is taken as the argument for the loss in eq. (30) instead of $a_5$.

As for the inputs, all networks are trained on functions of the four-momenta of the sampled points, as it is observed that training directly on the original parametrization presented in eq. (8) and eq. (22) yields subpar results (see Figure 16). Plus, L-GATr can only be trained on four-momenta inputs due to architecture constraints. We derive the four-momenta for each point from the sampled phase space parameters. In doing so, we prepare the $t\bar{t}$ inputs for $f_1$–$f_4$ so that the angle $\varphi_t$ lies only in the range $[0, \pi]$. This decision allows all neural networks to skip learning about the parity symmetry of these functions during training and reduces the interpolation domain (see option 3 from Section 2.6).

All networks are trained for $5 \times 10^5$ steps with a batch size of 256, the Adam optimizer [103], and a Cosine Annealing scheduler [104]. Higher amounts of training iterations were observed to produce better performance (see Table 3). However, we choose to refrain from increasing the training time due to prohibitive training times on L-GATr. Due to instabilities during training, we refrain from applying early stopping and we perform validation checks every 300 iterations.

## 6.1 MLP

The MLP is built as a simple fully connected neural network with GELU activation functions [105]. We investigate two versions of it: MLP($x$), which takes $\vec{x}$ as its inputs (the same as interpolation methods from the previous chapters), and MLP($s$), which takes the Lorentz invariants built from pairs of momenta,

$$s_{ij} \equiv (p_i + p_j)^2, \quad \text{with } i \neq j, \tag{62}$$

amounting to 10 inputs for $f_1$–$f_4$ and 6 inputs for $f_5$, as opposed to MLP($x$), which takes 5 and 2 correspondingly.

Both MLP architectures consists of 5 hidden layers with 512 hidden channels each for all test functions, amounting to $10^6$ learnable parameters. The inputs are preprocessed by taking

| Network layers | Hidden channels | Training iterations | Learning rate | $\epsilon$ |
|:---:|:---:|:---:|:---:|:---:|
| 3 | 512 | $5 \times 10^5$ | $10^{-3}$ | $4.93 \times 10^{-5}$ |
| 3 | 512 | $5 \times 10^5$ | $2 \times 10^{-4}$ | $4.81 \times 10^{-5}$ |
| 3 | 512 | $5 \times 10^5$ | $10^{-5}$ | $2.66 \times 10^{-4}$ |
| 3 | 512 | $5 \times 10^6$ | $10^{-3}$ | $2.44 \times 10^{-5}$ |
| 3 | 512 | $5 \times 10^6$ | $2 \times 10^{-4}$ | $1.50 \times 10^{-5}$ |
| 3 | 512 | $5 \times 10^6$ | $10^{-5}$ | $4.16 \times 10^{-5}$ |
| 5 | 512 | $5 \times 10^5$ | $10^{-3}$ | $5.90 \times 10^{-5}$ |
| **5** | **512** | $\mathbf{5 \times 10^5}$ | $\mathbf{2 \times 10^{-4}}$ | $\mathbf{4.37 \times 10^{-5}}$ |
| 5 | 512 | $5 \times 10^5$ | $10^{-5}$ | $2.60 \times 10^{-4}$ |
| 5 | 512 | $5 \times 10^6$ | $10^{-3}$ | $4.07 \times 10^{-5}$ |
| 5 | 512 | $5 \times 10^6$ | $2 \times 10^{-4}$ | $1.51 \times 10^{-5}$ |
| 5 | 512 | $5 \times 10^6$ | $10^{-5}$ | $2.47 \times 10^{-5}$ |
| 10 | 1024 | $5 \times 10^5$ | $10^{-3}$ | $9.72 \times 10^{-5}$ |
| 10 | 1024 | $5 \times 10^5$ | $2 \times 10^{-4}$ | $5.86 \times 10^{-5}$ |
| 10 | 1024 | $5 \times 10^5$ | $10^{-5}$ | $2.25 \times 10^{-4}$ |

Table 3: Summary of the MLP hyperparameter scan based on trainings with $10^6$ $f_1$ data points. We highlight the setup that is used for the test function studies.

their logarithms and performing standardization similar to the amplitude preprocessing in eq. (61). We fix the maximum learning rate at $2 \times 10^{-4}$ for $f_1$–$f_4$, and at $10^{-3}$ for $f_5$. An average training on $10^6$ points lasts around 40 minutes on an Nvidia A40 GPU. A scan over training runs with $10^6$ data points is performed to select the hyperparameters, which comprise both the network structure and the learning parameters. This scan is executed by selecting an initial set of hyperparameter values and then increasing and decreasing each one by factors of 2 and 10 iteratively until the best performance is reached. We have observed that moderate changes in the network shape on a fixed parameter budget have a very small effect on network performance for all test functions. A summary of all the tests performed for hyperparameter optimization is shown in Table 3.

## 6.2 L-GATr

*Equivariant neural networks* constitute a very attractive option for any problem where symmetries are well defined [99, 100, 106–108]. These networks respect the spacetime symmetry properties of the data in every operation they perform. They do so by imposing the equivariance condition, defined as

$$f(\Lambda(x)) = \Lambda(f(x)), \tag{63}$$

where $x$ is a network input, $f$ is a network operation and $\Lambda$ is a Lorentz transformation. By restricting the action of the network to equivariant maps, it does not need to learn the symmetry properties of the data during training and its range of operations gets reduced to only those allowed by the symmetry. This makes equivariant networks very efficient to train and capable of reaching high performance with low amounts of training data.

L-GATr is a neural network architecture that achieves equivariance by working in the spacetime geometric algebra representation [109]. A geometric algebra is generally defined as an extension of a vector space with an extra composition law: the geometric product. Given two vectors $x$ and $y$, their geometric product can be expressed as the sum of a symmetric and an

antisymmetric term

$$xy = \frac{\{x,y\}}{2} + \frac{[x,y]}{2}, \tag{64}$$

where the first term can be identified as the usual vector inner product and the second term represents a new operation called the *outer product*. The outer product allows for the combination of vectors to build higher-order geometric objects. In this particular case, $[x,y]$ is a bivector, which represents an element of the plane defined by the directions of $x$ and $y$.

The spacetime algebra $\mathbb{G}^{1,3}$ can be built by introducing the geometric product on the Minkowski vector space $\mathbb{R}^{1,3}$. The geometric product in this space is fully specified by demanding that the basis elements of the vector space $\gamma^\mu$ satisfy the following anti-commutation relation:

$$\{\gamma^\mu, \gamma^\nu\} = 2g^{\mu\nu}. \tag{65}$$

This inner product establishes that the vectors $\gamma^\mu$ in the context of the spacetime algebra have the same properties as the gamma matrices from the Dirac algebra. Both algebras are tightly connected, with the Dirac algebra representing a complexification of the spacetime algebra.

With this notion in mind, we can now cover all unique objects in the spacetime algebra by taking antisymmetric products of the gamma matrices. A generic object of the algebra is called a multivector, and it can be expressed as

$$x = x^S \, 1 + x^V_\mu \, \gamma^\mu + x^B_{\mu\nu} \, \sigma^{\mu\nu} + x^A_\mu \, \gamma^\mu\gamma^5 + x^P \, \gamma^5 \qquad \text{with} \qquad \begin{pmatrix} x^S \\ x^V_\mu \\ x^B_{\mu\nu} \\ x^A_\mu \\ x^P \end{pmatrix} \in \mathbb{R}^{16}. \tag{66}$$

In this expression, the components of the multivector are divided in grades, defined by the number of gamma matrix indices that are needed to express them. Namely, $x^S 1$ constitutes the scalar grade, $x^V_\mu \gamma^\mu$ the vector grade, $x^B_{\mu\nu}\sigma^{\mu\nu}$ the geometric bilinear grade, $x^A_\mu\gamma^\mu\gamma^5$ the axial vector grade, and $x^P\gamma^5$ the pseudoscalar grade.

Apart from its extended representation power, the spacetime algebra offers a clear way to define equivariant transformations with respect to the Lorentz group on a wide range of geometric objects in Minkowski space. The main consideration is that Lorentz transformations on algebra elements act separately on each grade [99, 110]. As a consequence, any equivariant map must transform all components of a single grade in the same manner and allow for different grades to transform independently.

This guideline allows for an easy adaptation of standard neural network layers to equivariant operations in the algebra. In the case of L-GATr, this adaptation is performed on a transformer backbone. *Transformer* [111] is a neural network architecture that is well suited to deal with datasets organised as sets of particles, since its attention mechanism can be leveraged to capture correlations between them in an accurate way. L-GATr is built with equivariant versions of linear, attention, normalisation, and activation layers [111, 112]. It also includes a new layer MLPBlock featuring the geometric product to further increase the expressivity of the network. Its layer structure is

$$\bar{x} = \text{LayerNorm}(x)$$
$$\text{AttentionBlock}(x) = \text{Linear} \circ \text{Attention}(\text{Linear}(\bar{x}), \text{Linear}(\bar{x}), \text{Linear}(\bar{x})) + x$$
$$\text{MLPBlock}(x) = \text{Linear} \circ \text{Activation} \circ \text{Linear} \circ \text{GP}(\text{Linear}(\bar{x}), \text{Linear}(\bar{x})) + x$$
$$\text{Block}(x) = \text{MLPBlock} \circ \text{AttentionBlock}(x)$$
$$\text{L-GATr}(x) = \text{Linear} \circ \text{Block} \circ \text{Block} \circ \cdots \circ \text{Block} \circ \text{Linear}(x). \tag{67}$$

All of these layers are redefined to operate on multi-vectors and restricted to act on algebra grades independently to ensure equivariance. Further details for each of the layers are provided in Refs. [99, 100].

Through this procedure, we build knowledge about the Lorentz symmetry into the architecture, but it can also be used to enforce awareness of the discrete symmetries described in Section 2.6 besides the parity invariance present in $f_1$–$f_4$ (i.e., option 1 from Section 2.6). Being a transformer, L-GATr handles individual particle inputs independently and can enforce any exchange symmetry on individual particles. This is performed in practice by including common scalar labels to every set of particles that leave the amplitude invariant under permutation.

To operate with L-GATr, we need to embed inputs into the geometric algebra representation and undo said embedding once we obtain the outputs. For the interpolation problem, inputs always consist of particle 4-momenta $p$, which can be embedded into multi-vectors as

$$x_\mu^V = p_\mu \qquad \text{and} \qquad x^S = x_{\mu\nu}^T = x_\mu^A = x^P = 0 \,. \tag{68}$$

Each particle is embedded to a different multivector with 16 components, resulting in the inputs being organized as sets of particles or tokens, 5 for $f_1$–$f_4$ and 4 for $f_5$. Each token also incorporates a distinct integer label as part of the inputs. The value for these labels is selected according to the permutation invariance pattern of each process we study. Specifically, we use the same token label for the particles in the initial state for all processes except $f_2$, and we also use a shared label for the $t$ and $\bar{t}$ tokens for $f_1$, $f_3$ and $f_4$. This ensures that L-GATr also respects any instance of particle exchange symmetry. The token labels are not part of the multi-vectors, they are fed to the network as separate scalar inputs. Multi-vectors and scalars traverse the network through parallel tracks, mixing with each other only at the linear layers.

The output of the network is constructed through the use of a global token, which is introduced as an extra empty particle entry. This extra token holds no meaning at the input level, but at the output level its scalar component represents the estimator for the target amplitude. As for the input preprocessing, in L-GATr we divide them by the standard deviation over each of the particle momenta to prevent violating equivariance.

Our L-GATr build consists of a network with 8 attention blocks, 64 multivector channels, 32 scalar channels and 8 attention heads for all test functions, resulting in $7 \times 10^6$ learnable parameters. We set the maximum learning rate to $2 \times 10^{-4}$ for $f_1$–$f_4$, and $5 \times 10^{-4}$ for $f_5$. An average L-GATr training on $10^6$ points lasts around 40 hours on an Nvidia A40 GPU. As with the MLP, these values are chosen after a scan on trainings with $10^6$ points to maximize performance on that data regime. We observe only a small effect in performance from changing the network shape on a fixed parameter budget across the different test functions. A summary of all the tests performed for hyperparameter optimization is shown in Table 4. The tests on this architecture are carried out using the same procedure as with the MLP, but they are more limited due to substantially longer training times.

## 6.3  Discussion

We show the results for the MLP and L-GATr networks in Figure 16, where we display the results from training and testing on unweighted samples. MLP($x$) performs consistently worse than MLP($s$). Both MLP($s$) and L-GATr display very similar performance for all test functions; MLP($s$) is slightly better for $f_5$, and $f_2$ in the low-precision limit, but slightly worse for $f_4$ in the high-precision limit.

We also observe a slowdown in improvement for all networks as we increase the training data. This pattern has been empirically noted before [113–117]: the test loss goes down with the number of training points as a power law, but can eventually reach a lower bound, past

| Blocks | Multivector channels | Scalar channels | Learning rate | $\varepsilon$ |
|--------|---------------------|-----------------|---------------|---------------|
| 4 | 32 | 16 | $10^{-3}$ | $7.70 \times 10^{-5}$ |
| 4 | 32 | 16 | $2 \times 10^{-4}$ | $1.07 \times 10^{-4}$ |
| 4 | 32 | 16 | $10^{-5}$ | $3.31 \times 10^{-4}$ |
| 4 | 64 | 32 | $10^{-3}$ | $8.79 \times 10^{-5}$ |
| 4 | 64 | 32 | $2 \times 10^{-4}$ | $6.44 \times 10^{-5}$ |
| 4 | 64 | 32 | $10^{-5}$ | $1.72 \times 10^{-4}$ |
| 8 | 64 | 32 | $10^{-3}$ | $6.99 \times 10^{-5}$ |
| **8** | **64** | **32** | $\mathbf{2 \times 10^{-4}}$ | $\mathbf{5.15 \times 10^{-5}}$ |
| 8 | 64 | 32 | $10^{-5}$ | $6.57 \times 10^{-5}$ |

Table 4: Summary of the L-GATr hyperparameter scan based on trainings with $10^6$ $f_1$ data points. All network builds are trained for $5 \times 10^5$ iterations. We highlight the setup that is used for the test function studies.

which the performance cannot improve even with infinite data. The causes for this may involve insufficient training iterations, noise in the data, and overfitting. In our case, since our data is fully deterministic and we observe no overfitting in the large dataset regime, we theorize that the main bottleneck is training time. Indeed, the left panel of Figure 17[4] and Table 3 show that longer trainings on the MLP substantially improves the performance in the large data limit. Despite these observations, a dedicated study to better understand this limitation in the context of amplitude interpolation may still be needed.

Comparing with other methods, machine learning algorithms surpass other interpolation methods in the low-to-medium precision regime for $f_1$–$f_4$, and they are only overtaken in the high precision limit, after machine learning results stop improving. The advantage over other approaches is the greatest for $f_2$ and $f_4$, which represent the more complicated higher order corrections to amplitudes. This signals potential utility of machine learning approaches for the interpolation of more complex multi-loop amplitudes.

The only front where neural networks underperform is in the case of $f_5$. We note that this is the only 2-dimensional function we test, and that seeing less benefits from neural networks in lower dimensions should not be surprising.

Another important result that we obtain from this study is that neural network training appears to be insensitive to the distribution of points it is trained on. As can be seen from the right panel of Figure 17, we can train the MLP on either uniform or unweighted samples and get similar performance for all training set sizes. This means that we can e.g. train our algorithms on naively sampled datasets, while targeting evaluation on physically motivated distributions without a significant performance degradation.

From Figure 18 we observe that dividing by the leading-order amplitudes $a_{\rm LO}$ brings a small but stable improvement for $f_2$ and $f_4$, unlike for polynomials and sparse grids, where the advantage disappears with more data.

Finally, we note that it should be possible to adaptively add to the training dataset focusing on regions that are poorly estimated in the initial stages of the training. This might work especially well combined with methods that provide uncertainty estimates on predictions, such as Bayesian neural networks [97,118–121] or repulsive ensembles [122,123]. We refrain from investigating such strategies in this paper, leaving them for future work.

---

[4]In this plot the degraded performance at the $10^3$–$10^4$ data point range is due to overfitting, which is more severe in this long training regime due to the slower learning rate reduction set up by the scheduler.

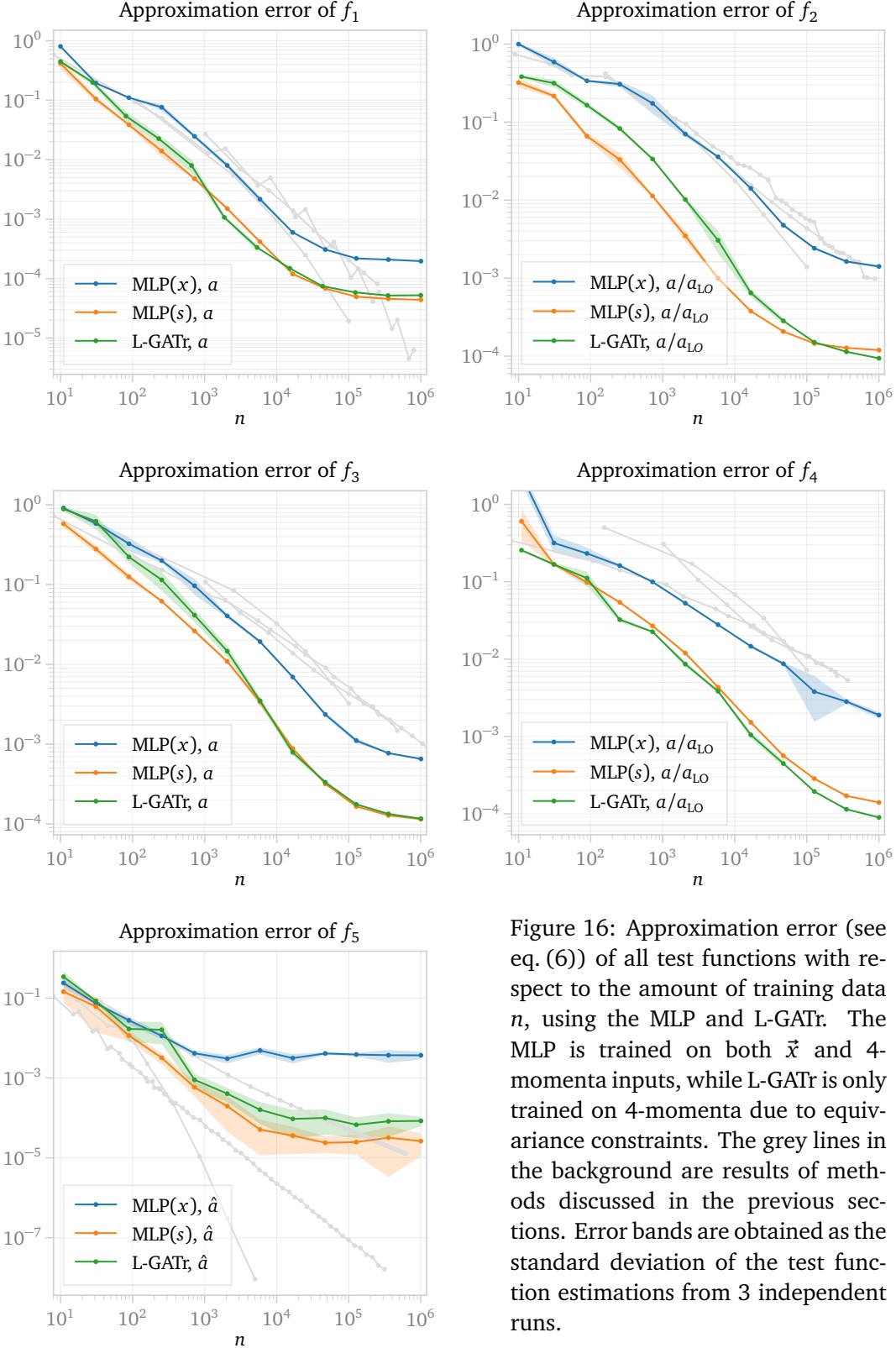

Figure 16: Approximation error (see eq. (6)) of all test functions with respect to the amount of training data $n$, using the MLP and L-GATr. The MLP is trained on both $\vec{x}$ and 4-momenta inputs, while L-GATr is only trained on 4-momenta due to equivariance constraints. The grey lines in the background are results of methods discussed in the previous sections. Error bands are obtained as the standard deviation of the test function estimations from 3 independent runs.

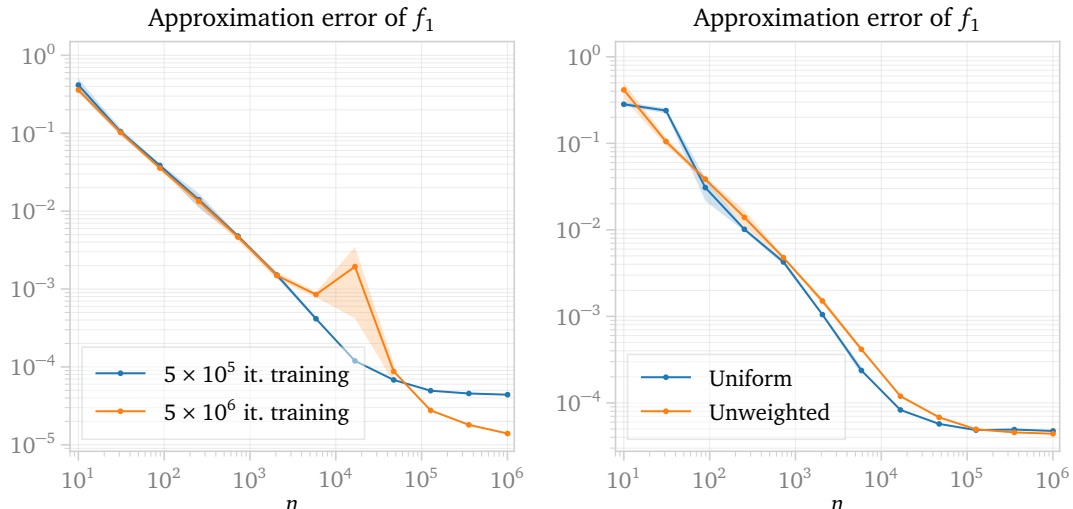

Figure 17: Comparison of the effect of training duration (left) and sampling method choice (right) on the $f_1$ MLP performance. Error bands are obtained as the standard deviation of the test function estimations from 3 independent runs.

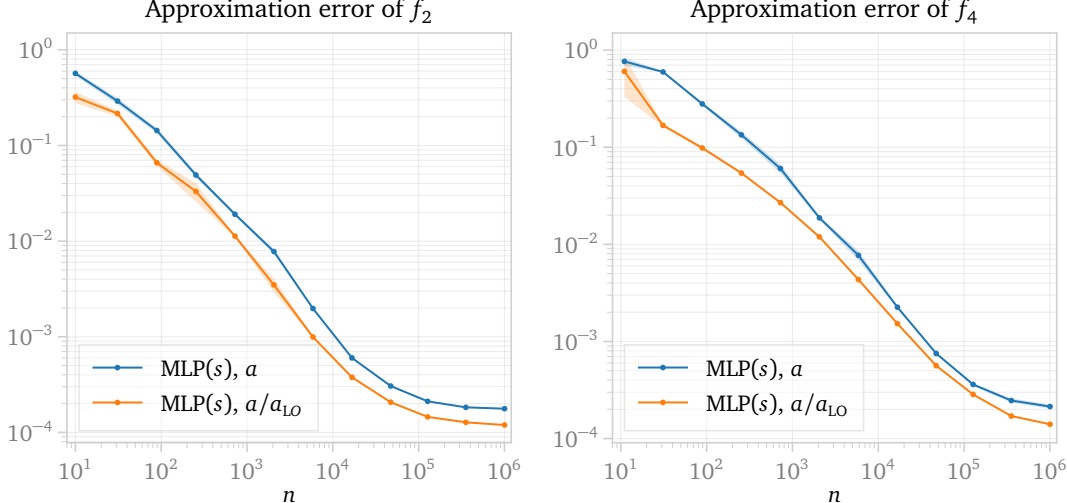

Figure 18: Comparison of $f_2$ (left) and $f_4$ (right) MLP trainings on the $a/a_{\mathrm{LO}}$ ratio versus the loop amplitudes $a$. Error bands are obtained as the standard deviation of the test function estimations from 3 independent runs.

# 7 Conclusions

We have presented a detailed investigation of several frameworks to interpolate multi-dimensional functions, focusing on scattering amplitudes depending on a number of phase-space variables. We have used amplitudes related to $t\bar{t}H$ production (5-dimensional phase space, test functions $f_1$–$f_4$) and Higgs+jet production (2-dimensional phase space, test function $f_5$) at the LHC as test functions, and applied various ways of polynomial interpolation, B-spline interpolation, spatially adaptive sparse grids, and interpolation based on machine learning techniques, using a standard multi-layer perceptron and the Lorentz-Equivariant Geometric Algebra Transformer.

Considering that the evaluation of loop amplitudes is costly, the main performance measure is how the approximation error of the different interpolation methods scales with the number of data points.

To measure the approximation error we notice that loop amplitudes typically have peaks or show a steep rise towards the phase-space boundaries. However, the contribution of such regions to physical observables might be small because it is suppressed by phase-space density or parton distributions. With this in mind, we have used physically motivated error metrics that weigh the approximation errors in certain phase-space regions proportionally to their contribution to the total cross section.

For the 5-dimensional test functions, we find that machine learning techniques significantly outperform the classical interpolation methods at $10^{-2}$ to $10^{-4}$ approximation error, in some cases by several orders of magnitude. For the lower dimensional case, we see that most methods perform well enough, but if very high precision is required, sparse grids are the best choice. In fact, for all test functions we see that the performance of machine learning approaches starts to stagnate after a certain precision threshold, while classical interpolation approaches scale smoothly. Since this threshold is fairly high, this still leaves machine learning techniques as the best practical choice in high dimensions, but does call for further investigation.

For test functions representing higher order correction to amplitudes, we see an improvement from training on the ratio $a/a_{\mathrm{LO}}$ instead of the raw loop amplitude $a$. For some methods the improvement is only significant in the low-data regime, while the neural network approaches benefit from it at all training sizes.

We did not include two-loop examples in this work because, for simple cases where training data can be generated easily (e.g. the production of a colourless boson or massless $2 \to 2$ scattering) the phase space is of low dimensionality, such that any interpolation method would work well. More useful two-loop examples would be drawn from $2 \to 3$ scattering, but generating training data for such two-loop amplitudes would go beyond the scope of this paper.

Another important measure for the usefulness of a particular method is its extendibility. Since it is difficult to predict how much data is needed to reach the desired error target, it is likely that more training data needs to be added between validations. Both the adaptive methods and neural networks provide this option. Spatially adaptive sparse grids are particularly flexible: they allow for the addition of single points at a time and for an assessment of which regions are approximated the worst, and thus need more points. Neural networks are fairly insensitive to the data point distribution, which allows for more data to be easily added, at the cost of a repeated training.

In summary, our investigations show that the construction of multi-dimensional amplitude interpolation frameworks with a sufficient precision for current and upcoming collider experiments is feasible, with machine learning being the most promising tool for it. Therefore, numerical calculations for multi-scale loop amplitudes continue to have a promising future.

Finally, we provide implementations of the described methods, as well as the test functions used for the benchmarks, at https://github.com/OlssonA/interpolating_amplitudes.

# Acknowledgements

We would like to thank Bakul Agarwal, Anja Butter, Benjamin Campillo, Tobias Jahnke, Stephen Jones, Matthias Kerner, and Jannis Lang for interesting discussions and/or contributions to the test functions. This research was supported by the Deutsche Forschungsgemeinschaft (DFG, German Research Foundation) under grant 396021762 - TRR 257. V.B. acknowledges financial support from the Grant No. ASFAE/2022/009 (Generalitat Valenciana and MCIN, NextGenerationEU PRTR-C17.I01).

# A    Additional details

## A.1    Phase space parametrisation for $pp \to t\bar{t}H$

The variables used for test functions $f_1$ to $f_4$ originate from a $2 \to 3$ phase space to produce a top quark pair and a Higgs boson, where we use three angular variables and two energy variables:

$$d\Phi_{t\bar{t}H} = \frac{1}{2^{10}\pi^4 \hat{s} s_{t\bar{t}}} \sqrt{\lambda(s_{t\bar{t}}, m_t^2, m_t^2)} \sqrt{\lambda(\hat{s}, s_{t\bar{t}}, m_H^2)} \times$$

$$\Theta(\sqrt{\hat{s}} - 2m_t - m_H)\Theta(s_{t\bar{t}} - 4m_t^2)\Theta([\sqrt{\hat{s}} - m_H]^2 - s_{t\bar{t}})\,ds_{t\bar{t}}\,d\Omega_{t\bar{t}} \sin\theta_H\,d\theta_H \quad (69)$$

with $\theta_H$ being the polar angle of the Higgs boson relative to the beam axis, $d\Omega_{t\bar{t}} = \sin\theta_t\,d\theta_t\,d\varphi_t$ and $\lambda$ being the Källén function, $\lambda(a, b, c) = a^2 + b^2 + c^2 - 2ab - 2bc - 2ca$, depending on the variables

$$\hat{s} = (p_3 + p_4 + p_5)^2 \,,\; s_{t\bar{t}} = (p_3 + p_4)^2 \quad (70)$$

and the masses. As the production threshold of the $t\bar{t}H$ system is located at $s_0 = (2m_t + m_H)^2$, a convenient variable for a scan in partonic energy is

$$\beta^2 = 1 - \frac{s_0}{\hat{s}} \quad (71)$$

such that $\beta^2 = 0$ at the production threshold and $\beta^2 \to 1$ in the high energy limit. For more details we refer to Ref. [37].

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
