# Peer review of "Interpolating amplitudes"

_SciPost Physics, doi:SciPost Phys. 19, 123 (2025)_

## Round 1 · Referee Report · Anonymous (Referee 1) · 2025-3-27

Strengths

1- The authors demonstrate the ability to leverage interpolation techniques to speed up scattering amplitude calculations 2- The authors perform a detailed analysis of various methods 3- The authors provide their code, improving the reproducibility of their work

Weaknesses

1- The authors neglect a large swath of literature that focuses on improving computational costs for event generation. 2- The authors have written the paper trying to balance between explaining the physics to mathematicians and the math to physicists. However, this balance does not work well in the introduction. They over simplify both sides of the discussion too far to make it clear what is going on.

Report

The work of the authors is interesting, and an important development towards making efficient high-precision predictions including higher order corrections. While the work is interesting and an important contribution to the field, it is not suitable for publication in its current form. The authors neglect a large swath of other research on improving the computational efficiency of event generators, do not discuss where the most significant computational bottlenecks arise from in the calculations, nor do they test out any two-loop processes in which this technique would prove to be of most use. Below are a list of requested changes to improve the overall quality and presentation of the work.

Requested changes

1- In the abstract, the authors talk about the dimensionality of the interpolation problems. However, a reader may confuse this with techniques such as dimensional regularization used for evaluating loop integrals, since the discussion of multi-loop amplitudes features prominently in the previous sentence. The authors should make the language more precise. 2- Also in the abstract, the authors state that "This work aims to fill this gap...", but it is unclear what they mean by the gap. The gap in efficient interpolation frameworks? The gap in numerical evaluations of amplitudes? 3- The authors state in the abstract that they are reviewing interpolation methods and "assessing their performance for concrete examples drawn from particle physics." However, the whole first paragraph focuses solely on problems related to particle physics, so here mentioning this seems out of place since the authors have only talked about particle physics thus far. 4- In the introduction, there are exactly 3 references which are just reviews of the progress in higher loop calculations. There are many claims throughout the introduction that should be supported with appropriate references: 4a- The first sentence claims about the precision needs of the experiments for BSM searches. There should be some general references to current collider papers from CMS and ATLAS highlighting their needs. 4b- The authors point out that "two-loop amplitudes become rapidly unfeasible as the number of external legs..." but provide no reference to any of the work that demonstrates this. 4c- The authors claim that numerical methods are more tractable, but again without any reference to works proving this. 4d- The authors talk about the traditional pipeline for Monte Carlo event generation in which millions of phase-space points need to be evaluated, but neglect a large collection of work in this area investigating computational costs, where the bottlenecks are, and attempts to address them. These would include papers such as: 2004.13687 (and references within), 2209.00843, 2203.11110 (and references within), 2112.09588, 1905.05120, etc. 4e- The authors also don't seem to recognize that in modern event generators, most of the computational time is spent on phase space generation, tree-level diagrams, real corrections, and double real corrections (see 2209.00843, talk by Simone Alioli at the "Event generators' and N(n)LO codes' acceleration" workshop held at CERN in November 2023, 2407.02194 (abstract mentions that double-reals are the most computationally intensive part), etc). 4f- To the end above, the authors neglect all efforts in improving the computational efficiencies of the base calculations, such as working on implementing the evaluation of amplitudes on GPUs (2106.06507,1305.0708,0908.4403,0909.5257,2303.18244, 2105.10529,2106.10279,2106.12631,2311.06198,2502.07060,2503.07439), improving phase space handling (1810.11509, 2001.05478, 2001.05486, 2001.10028, 2212.06172, 2211.02834, 2205,01697, 2302.10449, 2311.01548, 2401.09069, 2408.01486, etc.), reducing negative weights (2109.07851,2303.15246,2005.09375,2007.11586,2002.12716,2411.11651,etc.), etc. 5- In the section "What is an amplitude," the authors introduce the calculations of the perturbative series, but focus solely on the loop corrections and completely neglect any discussion of the real corrections that are also required. 6- The authors set the target goal to be at 1%, but provide no justification for this number, and if it is sufficient for the HL-LHC. This is at least an order of magnitude too small for the processes they consider in their tests. 7- The authors discuss how their work would provide a low precision result in the "very-high-energy region" where the contribution to the total cross section is small. However, in the introduction the authors state that precision is needed for BSM, and the effects of BSM will mostly show up in the tails of the distributions and not in the peaks. The precision in the peaks will be needed for electroweak precision tests, Higgs measurements, etc. but not general BSM searches. 8- The authors point out that normalizing flows can be used for find the variable transformation, and simply cite a review article on normalizing flows and neglect the work done by the particle physics community to use this technique. 9- In the test function section, the authors use the language "Leading-order (tree-level) amplitudes", but do not clearly define this. If the goal is to obtain help from mathematicians, these terms should be more well defined. 10- The authors show projections of the test functions, but these are not labeled as figures. These should be figures with captions and not inline plots. 11- The authors use the phrase "integrable Coulomb-type singularity" which should be better defined for the intended mathematician audience. 12- In the introduction, the authors discuss how the two-loop amplitudes really are a challenge as motivation for their study. It would be very useful for them to include some simple two-loop example as a test function to demonstrate that their approach will work for two loops and not limited to tree and one-loop amplitudes. 13- The authors claim that the RAMBO sampling technique is "widely" used. This is not the case in most generators any more. The authors should remove the "widely" and supply references to more modern phase space techniques such as the multichannel approaches, diagram based approaches (MadGraph), recursion-based approaches (COMIX), etc. 14- In all the figures, the y-axes have no labels. The y-axis should be labeled to make it more clear what is shown. 15- In figure 2, the authors should add x- and y-axes labels to all plots. 16- In figure 4, the authors mention that n is the number of evaluations, but do not mention this in the other figures. The authors should add what n is in the caption of the other figures for clarity. 17- Section 3.8 is dedicated to discussing the impact of noise on the interpolation, but do not motivate why there would be such low precision values of a. A discussion of this should be included. 18- Figure 5 is mentioned before figure 4. The order should be switched as is standard requirements for publications 19- At the end of section 5.3, the authors discuss the difference between a greedy approach and a balanced approach as show in Figure 10, but do not explain what function was used in this demonstration. It is mentioned in the caption, but should be mentioned in the main text as well. 20- In section 5.5, the authors discuss that greedy refinement is "not very robust against numerical artefacts and noise." What do the authors mean here by numerical artefacts? 21- The sentence "We see in each test function that the scaling is better, however, and that at high amounts of training data the improvements are more significant" is not clear at all. The authors should rework this sentence. 22- In the discussion of machine learning techniques, the authors talk about the importance of enforcing Lorentz invariance in their network, but do not discuss the importance of enforcing the correct QCD soft and collinear structure. They cite the paper that demonstrated the importance of this as Ref. 57, but do not discuss the novelty of this work. 23- In the sentence "... we can train the MLP on either uniform or unweighted samples and the get similar performance...", the "the" should be dropped after the "and". 24- Overall, the paper could use a read through for clarity and grammar.

Recommendation

Ask for major revision

---

## Round 1 · Referee Report · Anonymous (Referee 2) · 2025-4-1

Strengths

  • Self-contained review of interpolation methods for the approximation of higher-order loops scattering amplitudes with a detailed discussion of the benefits and pitfalls of all the interpolation techniques;
  • Extensive exploration of the configurations used to define the interpolation functions and the quantity of interest, either $a$ or $f$;
  • The work is timely and addresses important questions for future precision measurements;
  • The manuscript could be a common ground for the collaboration between physicists and mathematicians.

Weaknesses

  • The methods themselves are not novel. The splines and the polynomial methods seem to be already well-established, and the machine learning approaches have also been applied to NLO and high-multiplicity processes;
  • Evaluation speed is a recurrent theme throughout the paper, however, no numerical comparison between the methods has been made;
  • The comparison of machine learning methods is rather limited;
  • The datasets of all the test functions could be published. It will improve the reproducibility and future developments.

Report

The work proposed in the paper is valid, and the authors show a great understanding of the numerical tools. It is a timely study, which clarifies the direction for future research on amplitude regression. Some clarifications are needed, in particular, the introduction should be supported by references. The authors refer to two-loop amplitudes as problematic but only test the numerical tool up to one-loop contributions. The studies in the machine learning section are minimal compared to the other methods. Considering that the authors conclude that machine learning is the most promising tool, the studies could be extended.
Before recommending publication, these points should be addressed. Please refer to the requested changes for a detailed list.

Requested changes

  1. The introduction should be extended with references to support the authors' claims. For instance, in the first two paragraphs, the authors refer to the need for precision for future BSM searches and the difficulties of performing higher-order analytic calculations. Similarly, "... the test functions used in mathematics often have different properties than the functions related to loop amplitudes" should be clarified and referenced;
  2. In Sec.2.3 the authors introduce a target $\epsilon \sim 1\%$. How do the authors decide on this target? How does it would change with different evaluation metrics?
  3. In the same section, I think the symbol $\sigma_R$ is not introduced; 4a. The purpose of the slices in $x$ shown in Sec.2.5 is unclear since it is limited to only a few variables and does not cover the full space. The authors could consider reducing the number of slices and highlight important aspects of the amplitudes, e.g. divergent regions or symmetries; 4b. The formatting could be improved by including the slices as figures, which should also ease the discussion; 5a. The distinction between sections in Sec.3 seems overdone. Is there a motivation for dividing the discussion on Chebyshev polynomials in the three short subsections 3.1, 3.2, and 3.3? The authors could consider merging them into a single subsection; 5b. Similar argument applies to 3.8,3.9;
  4. Eq.(36), $\text{d}x$ is missing;
  5. What is the origin of the checkerboard pattern in Fig.3?
  6. The order of Fig.4 and Fig.5 should be swapped since the noise is discussed first.
  7. The motivation and the discussion on the effects of noise should be extended. The noise indeed acts as a lower bound for $\epsilon$ but it also worsens the scaling of the methods with the number of amplitudes, as can be seen from Fig.5. This potentially undermines the main scope of the numerical technique and should be discussed.
  8. At the end of page 17 there is a typo: was -> way;
  9. At the end of Sec.4.3 the authors introduce the advantageous evaluation time of the B-splines. However, no numerical evidence is shown. This is marginally important for this specific evaluation since the scaling is different but it is relevant given one of the main motivations of the paper, e.g. is the evaluation with a machine learning model faster than B-splines? The authors should consider introducing the evaluation time for all the methods studied.
  10. In the machine learning section, the authors move from the parameterization introduced in Sec.2.5 to the 4-momenta of the particles. Could the authors comment on this choice or show evidence of the superior performance of the proposed methods?
  11. The LGATR network is seven times bigger than the MLP. How was the choice of hyperparameters done for the two networks? Is this a "fair" comparison? The authors could extend the discussion with the following: a. Results with varying number of layers; b. Evaluation time on CPU and GPU; c. Comparison at fixed budget, e.g. the number of parameters, training time, or number of computations.

Recommendation

Ask for minor revision

---

## Round 2 · Referee Report · Anonymous (Referee 1) · 2025-8-25

Report

The authors have sufficiently addressed all my concerns. It is suitable for publication.

Recommendation

Publish (easily meets expectations and criteria for this Journal; among top 50%)

---

## Round 2 · Referee Report · Anonymous (Referee 2) · 2025-9-7

Report

The authors addressed all of my concerns. However, I still think that the paper is a missed opportunity to compare interpolation techniques on two-loop amplitudes, or just on a partial set, which is even the motivation for this work.
Such a comparison would provide a link between the already applied numerical interpolation techniques and the emerging machine learning tools.
In its current form, the paper is a comprehensive, much-needed, and publication-worthy collection of methodologies, which, however, falls short in applying them to the problem with the complexity of interest.

Small detail: Table 3 uses $\epsilon$ while the rest of the paper uses $\varepsilon$.

Recommendation

Publish (meets expectations and criteria for this Journal)

---

## Round 2 · Author Response

We thank the referees for their time, careful consideration, and the evaluation of our manuscript. We list below the changes we have made concerning the helpful suggestions.

Report #1:

1- In the abstract, the authors talk about the dimensionality of the interpolation problems. However, a reader may confuse this with techniques such as dimensional regularization used for evaluating loop integrals, since the discussion of multi-loop amplitudes features prominently in the previous sentence. The authors should make the language more precise.

We have updated the abstract to reduce the possible confusion, now writing explicitly "amplitude interpolation with more variables".

2- Also in the abstract, the authors state that "This work aims to fill this gap...", but it is unclear what they mean by the gap. The gap in efficient interpolation frameworks? The gap in numerical evaluations of amplitudes?

We have reworded this part of the abstract, in particular we do not use the term "gap" anymore.

3- The authors state in the abstract that they are reviewing interpolation methods and "assessing their performance for concrete examples drawn from particle physics." However, the whole first paragraph focuses solely on problems related to particle physics, so here mentioning this seems out of place since the authors have only talked about particle physics thus far.

We have reworded this part of the abstract, saying now explicitly that we use 2 -> 3 scattering amplitudes.

4- In the introduction, there are exactly 3 references which are just reviews of the progress in higher loop calculations. There are many claims throughout the introduction that should be supported with appropriate references:

4a- The first sentence claims about the precision needs of the experiments for BSM searches. There should be some general references to current collider papers from CMS and ATLAS highlighting their needs.

We have added four references relating to the need for more precision from the theory side for the LHC, the HL-LHC and future colliders that are discussed in the framework of the European Strategy for Particle Physics Update.

4b- The authors point out that "two-loop amplitudes become rapidly unfeasible as the number of external legs..." but provide no reference to any of the work that demonstrates this.

4c- The authors claim that numerical methods are more tractable, but again without any reference to works proving this.

We have added references [8-11] to address 4b and 4c. In addition, we point the readers to section 3 in Ref.[5] where these issues are discussed in detail.

4d- The authors talk about the traditional pipeline for Monte Carlo event generation in which millions of phase-space points need to be evaluated, but neglect a large collection of work in this area investigating computational costs, where the bottlenecks are, and attempts to address them. These would include papers such as: 2004.13687 (and references within), 2209.00843, 2203.11110 (and references within), 2112.09588, 1905.05120, etc.

We acknowledge that increasing the efficiency of event generation is a vibrant field of study. Our research focus, however, is very different. In our view, any form of event generation is incompatible with starting a separate, lengthy (two-)loop calculation for each individual event. Our aim is to replace this lengthy calculation (which might take hours of time and require specialized hardware) with a fast interpolation (taking under a second). For this aim, the details of the event generation do not matter, only the big picture.

We have modified the abstract and the introduction to make our aims clearer in this regard. We have also included references to 2004.13687 and 2203.11110 for the reader who wants to enter more deeply into the subject of event generation.

4e- The authors also don't seem to recognize that in modern event generators, most of the computational time is spent on phase space generation, tree-level diagrams, real corrections, and double real corrections (see 2209.00843, talk by Simone Alioli at the "Event generators' and N(n)LO codes' acceleration" workshop held at CERN in November 2023, 2407.02194 (abstract mentions that double-reals are the most computationally intensive part), etc).

This is true only for processes where the two-loop amplitudes are known in analytic form, as it is the case for e+e- to 3 jets at NNLO discussed in 2407.02194. Once the virtual amplitudes cannot be coded up analytically, the budget of computational time shifts dramatically towards the virtual amplitude, because numerical loop integrations would need to be done for each phase space point. This is exactly the problem we want to address in the manuscript, by providing the results in the form of a grid and an interpolation framework.

We have added a paragraph about the real radiation in section 2.1.

4f- To the end above, the authors neglect all efforts in improving the computational efficiencies of the base calculations, such as working on implementing the evaluation of amplitudes on GPUs (2106.06507,1305.0708,0908.4403,0909.5257,2303.18244, 2105.10529,2106.10279,2106.12631,2311.06198,2502.07060,2503.07439), improving phase space handling (1810.11509, 2001.05478, 2001.05486, 2001.10028, 2212.06172, 2211.02834, 2205,01697, 2302.10449, 2311.01548, 2401.09069, 2408.01486, etc.), reducing negative weights (2109.07851,2303.15246,2005.09375,2007.11586,2002.12716,2411.11651,etc.), etc.

Again, we would like to emphasize that our focus is very different. While the above references aim to accelerate tree- or NLO amplitude calculations, our aim is to make two-loop amplitudes that are not available analytically accessible for phenomenology. We have modified the introduction to clarify this point.

Similarly, the elimination of negative weights is an important topic for event generators, but it is orthogonal to the aims of this paper.

5- In the section "What is an amplitude," the authors introduce the calculations of the perturbative series, but focus solely on the loop corrections and completely neglect any discussion of the real corrections that are also required.

We have included a paragraph discussing real corrections and the conditions under which they are the slowest part to compute in Monte Carlo calculations of cross sections. We would like to note, however, that the reader does not need to understand the details of real radiation integration to understand this paper, as we are concerned with virtual amplitudes.

6- The authors set the target goal to be at 1%, but provide no justification for this number, and if it is sufficient for the HL-LHC. This is at least an order of magnitude too small for the processes they consider in their tests.

We have dropped the 1% target from the text of the paper, as it is not essential for our conclusions.

We note that the experimental precision on the example processes we consider (ttH production), is ~10% at the moment, and is projected to be ~2% at the end of HL-LHC. Therefore, knowing the 2-loop virtual corrections with 1% uncertainty should be sufficient for this process, considering that the NNLO correction is itself a small part of the whole cross-section, and that the scale uncertainties at NNLO are of order 3%.

7- The authors discuss how their work would provide a low precision result in the "very-high-energy region" where the contribution to the total cross section is small. However, in the introduction the authors state that precision is needed for BSM, and the effects of BSM will mostly show up in the tails of the distributions and not in the peaks. The precision in the peaks will be needed for electroweak precision tests, Higgs measurements, etc. but not general BSM searches.

We have added sentences pointing to the tradeoff that is present whenever an error target is chosen to section 2.3. We also discussed an alternative definition for the error target that would ensure a better precision guarantee for the tails of distributions, at the expense of requiring more data.

8- The authors point out that normalizing flows can be used for find the variable transformation, and simply cite a review article on normalizing flows and neglect the work done by the particle physics community to use this technique.

We have added a short mention of the usage of normalizing flows for importance sampling, and more references.

9- In the test function section, the authors use the language "Leading-order (tree-level) amplitudes", but do not clearly define this. If the goal is to obtain help from mathematicians, these terms should be more well defined.

We now explain these terms in the beginning, after eq.(5).

10- The authors show projections of the test functions, but these are not labeled as figures. These should be figures with captions and not inline plots.

The projections are included for illustration to provide the reader with an intuition of the general behaviour and the smoothness of the functions we are working with. We want to minimize how much these illustrations obstruct the reading process, therefore we decided to make them small, minimize the labels to the bare minimum, skip any legends, and format them inline. As we think this form is better for the logic flow the reader is following, we are reluctant to change this.

11- The authors use the phrase "integrable Coulomb-type singularity" which should be better defined for the intended mathematician audience.

We have rewritten this sentence to be more precise.

12- In the introduction, the authors discuss how the two-loop amplitudes really are a challenge as motivation for their study. It would be very useful for them to include some simple two-loop example as a test function to demonstrate that their approach will work for two loops and not limited to tree and one-loop amplitudes.

We agree, it would be useful. The problem is that simple two-loop examples, where training data can be generated easily (e.g. Drell-Yan or massless 2->2 scattering), are of low dimensionality, such that any interpolation method would work well. More realistic 2-loop examples would be drawn from 2->3 scattering, but generating training data for those would constitute a major calculation by itself, and go beyond the scope of this paper.

We have added a corresponding statement to the conclusions.

13- The authors claim that the RAMBO sampling technique is "widely" used. This is not the case in most generators any more. The authors should remove the "widely" and supply references to more modern phase space techniques such as the multichannel approaches, diagram based approaches (MadGraph), recursion-based approaches (COMIX), etc.

We have removed the "widely used" qualifier, as it is not important for the discussion.

14- In all the figures, the y-axes have no labels. The y-axis should be labeled to make it more clear what is shown.

Our data plots are fairly dense, and we prefer to use all the available horizontal space for the data, rather than the frame labels. For this reason the vertical axes are all labelled on top of the each plot (and additionally in the caption), rather than on their sides.

15- In figure 2, the authors should add x- and y-axes labels to all plots.

All the horizontal labels in Figure 2 would read "i", and all vertical ones would read "j". Instead of repeating these labels 10 times, we prefer to explain this in the caption. Otherwise the figure becomes even more crowded and hard to read than it already is.

16- In figure 4, the authors mention that n is the number of evaluations, but do not mention this in the other figures. The authors should add what n is in the caption of the other figures for clarity.

We have modified all figure captions to reflect this.

17- Section 3.8 is dedicated to discussing the impact of noise on the interpolation, but do not motivate why there would be such low precision values of a. A discussion of this should be included.

The motivation is that lower precision results are faster to obtain with any numeric loop integral evaluation method. The specific values of the noise parameter \sigma used in the plots, which are indeed very high, are chosen to amplify the effect of noise. We have added a sentence explaining this reasoning.

18- Figure 5 is mentioned before figure 4. The order should be switched as is standard requirements for publications

We have moved the first reference to figure 4 into an earlier section.

19- At the end of section 5.3, the authors discuss the difference between a greedy approach and a balanced approach as show in Figure 10, but do not explain what function was used in this demonstration. It is mentioned in the caption, but should be mentioned in the main text as well.

It is now also mentioned in the main text which test function was used for the comparison.

20- In section 5.5, the authors discuss that greedy refinement is "not very robust against numerical artefacts and noise." What do the authors mean here by numerical artefacts?

"Numerical artefacts" was a reference the possibility of receiving incorrect data from the external amplitude evaluations (which we have encountered in practice). We have removed the reference to these, since such problems are best handled separately. We keep the discussion about noise only.

21- The sentence "We see in each test function that the scaling is better, however, and that at high amounts of training data the improvements are more significant" is not clear at all. The authors should rework this sentence.

We have reworked this sentence.

22- In the discussion of machine learning techniques, the authors talk about the importance of enforcing Lorentz invariance in their network, but do not discuss the importance of enforcing the correct QCD soft and collinear structure. They cite the paper that demonstrated the importance of this as Ref. 57, but do not discuss the novelty of this work.

We have included a short sentence addressing this reference in more detail.

23- In the sentence "... we can train the MLP on either uniform or unweighted samples and the get similar performance...", the "the" should be dropped after the "and".

We have fixed the typo.

24- Overall, the paper could use a read through for clarity and grammar.

We have gone through the paper several times and made an effort to fix grammar mistakes and improve readability.

Report #2:

  1. The introduction should be extended with references to support the authors' claims. For instance, in the first two paragraphs, the authors refer to the need for precision for future BSM searches and the difficulties of performing higher-order analytic calculations. Similarly, "... the test functions used in mathematics often have different properties than the functions related to loop amplitudes" should be clarified and referenced;

We have extended the amount of references in the introduction. We have also listed the function spaces commonly employed in mathematical literature explicitly.

  1. In Sec.2.3 the authors introduce a target ϵ∼1%. How do the authors decide on this target? How does it would change with different evaluation metrics?

We have dropped the 1% target from the text of the paper, as it is not essential for our conclusions. We expect any practical application of these methods to select an error target appropriate for their goals. The motivation behind the original target was that many cross sections measured at LHC are known with >1% experimental uncertainty, such that having a 1% theoretical uncertainty on the best available prediction should be a good target for phenomenological purposes.

  1. In the same section, I think the symbol σR is not introduced;

\sigma_R refers to the cross-section integrated over the region R. We have clarified it as an explicit integral.

4a. The purpose of the slices in x shown in Sec.2.5 is unclear since it is limited to only a few variables and does not cover the full space. The authors could consider reducing the number of slices and highlight important aspects of the amplitudes, e.g. divergent regions or symmetries;

The slices are provided for illustration of the general behaviour and smoothness of the functions we investigate, so that the reader could immediately obtain a visual idea of what we are working with. We want to minimize how much these illustrations obstruct the reading process, therefore we decided to make them small, minimize the labels to the bare minimum, skip any legends, and format them inline. As we think this form is better for the logic flow the reader is following, we are very reluctant to change this.

We prefer to keep the discussion of the divergent regions and symmetries to text rather than image, because the divergent regions are in the end subtracted, so none of the functions are divergent. The full symmetries are too complicated to be easily shown, this would increase the number of slices.

4b. The formatting could be improved by including the slices as figures, which should also ease the discussion;

As explained, we prefer to keep these as small illustrations, not as full data plots.

5a. The distinction between sections in Sec.3 seems overdone. Is there a motivation for dividing the discussion on Chebyshev polynomials in the three short subsections 3.1, 3.2, and 3.3? The authors could consider merging them into a single subsection;

We have removed the numbering (but not the titles) from some of the smaller subsections. Our intention is that the subsection titles may guide the reader and therefore help with clarity.

5b. Similar argument applies to 3.8,3.9;

  1. Eq.(36), dx is missing;

We have fixed this.

  1. What is the origin of the checkerboard pattern in Fig.3?

Symmetries of the function. We have added a corresponding comment in the text.

  1. The order of Fig.4 and Fig.5 should be swapped since the noise is discussed first.

We have moved the first reference to figure 4 into an earlier section.

  1. The motivation and the discussion on the effects of noise should be extended. The noise indeed acts as a lower bound for ϵ but it also worsens the scaling of the methods with the number of amplitudes, as can be seen from Fig.5. This potentially undermines the main scope of the numerical technique and should be discussed.

Our interpretation of Figure 5 is that the scaling is not affected by the noise, except at the approximation errors close to the lower bound. We have now added two more noise levels to Figure 5: 3e-3 and 1e-3; they appear to support this conclusion.

  1. At the end of page 17 there is a typo: was -> way;

We have fixed this.

  1. At the end of Sec.4.3 the authors introduce the advantageous evaluation time of the B-splines. However, no numerical evidence is shown. This is marginally important for this specific evaluation since the scaling is different but it is relevant given one of the main motivations of the paper, e.g. is the evaluation with a machine learning model faster than B-splines? The authors should consider introducing the evaluation time for all the methods studied.

The exact evaluation time is highly dependent on the implementation, which is why we do not provide benchmarks for it in this paper. All methods in the paper evaluate fast enough compared to the time it takes to sample real corrections in MC programs.

  1. In the machine learning section, the authors move from the parameterization introduced in Sec.2.5 to the 4-momenta of the particles. Could the authors comment on this choice or show evidence of the superior performance of the proposed methods?

We have included several sentences addressing this point. We have also updated the plots with the results from training on the original parametrization to show the advantage of training the MLP with Lorentz invariants.

  1. The LGATR network is seven times bigger than the MLP. How was the choice of hyperparameters done for the two networks? Is this a "fair" comparison? The authors could extend the discussion with the following: a. Results with varying number of layers; b. Evaluation time on CPU and GPU; c. Comparison at fixed budget, e.g. the number of parameters, training time, or number of computations.

We have included a short description of our hyperparameter search, and added summary tables comparing different configurations we have investigated.

Our comparisons across all methods are fair in the fixed budget of the amount of data points used. Indeed, this fairness requirement is the primary goal of our investigation. We select the networks size, architecture, and input format to give the best results under it. Comparing under other budget constraints is interesting, and deserves a dedicated study, but we do not envision doing so in this paper.

The only other budget constraint we use is the training time of the networks, which may grow large, and we put a cap on it by fixing the number of training iterations (as mentioned in the paper).

We hope that with these changes, our article can now be accepted for publication in its present form.

Sincerely, V. Bresó-Pla, G. Heinrich, V. Magerya, A. Olsson

---

## Round 2 · List of Changes

In addition to the changes listed above, we have added a new
section 3.4, discussing polynomial interpolation methods not
based on the tensor-product grid, and benchmarking one recently
published instance of such methods. (Our conclusions remain
unchanged with the inclusion of this method).
We have also fixed typos in eq.(14) and eq.(19): the "-" there
should be "+" instead.

---

## Editorial Decision

published